# Integrative analysis reveals a conserved role for the amyloid precursor protein in proteostasis during aging

Vanitha Nithianandam[1,2], Hassan Bukhari[1,2], Matthew J. Leventhal[3,4], Rachel A. Battaglia[1,2], Xianjun Dong[2,5], Ernest Fraenkel[3] & Mel B. Feany[1,2] ✉

Aβ peptides derived from the amyloid precursor protein (APP) have been strongly implicated in the pathogenesis of Alzheimer's disease. However, the normal function of APP and the importance of that role in neurodegenerative disease is less clear. We recover the *Drosophila* ortholog of APP, Appl, in an unbiased forward genetic screen for neurodegeneration mutants. We perform comprehensive single cell transcriptional and proteomic studies of Appl mutant flies to investigate Appl function in the aging brain. We find an unexpected role for Appl in control of multiple cellular pathways, including translation, mitochondrial function, nucleic acid and lipid metabolism, cellular signaling and proteostasis. We mechanistically define a role for Appl in regulating autophagy through TGFβ signaling and document the broader relevance of our findings using mouse genetic, human iPSC and in vivo tauopathy models. Our results demonstrate a conserved role for APP in controlling age-dependent proteostasis with plausible relevance to Alzheimer's disease.

The amyloid precursor protein (APP) is a single transmembrane protein with large extracellular and small intracellular domains. Cleavage of APP within the extracellular region by the β-secretase protease and in the transmembrane domain by the γ-secretase complex results in the production of approximately 40 amino acid peptides, which aggregate into the extracellular plaques required for the diagnosis of Alzheimer's disease. Significant evidence implicates APP cleavage in Alzheimer's disease pathogenesis. Familial mutations in APP cause highly penetrant, autosomal dominant forms of Alzheimer's disease. Moreover, these mutations cluster around the β-secretase, and especially γ-secretase, cleavage sites. Mutations in the genes encoding presenilins, the γ-secretase complex member that performs the intramembrane cleavage liberating Aβ peptides, also cause highly penetrant forms of Alzheimer's disease. The functional effects of APP and presenilin mutations are somewhat varied, but the significant evidence suggests that disease-associated mutations favor the formation of longer, more aggregation prone Aβ42-43 species at the expense of

shorter, less amyloidogenic, Aβ40 peptides[1]. Remarkably, a rare variant of APP discovered in the Icelandic population and located near the β-secretase cleavage protects from the development of Alzheimer's disease and apparently from age-related cognitive decline as well[2].

These observations have focused significant efforts on the role of Aβ in Alzheimer's disease pathogenesis, with relatively less attention given to investigating the normal function of APP. Determining the endogenous function of APP has also been complicated experimentally by the presence of three closely related APP family members in vertebrates: APP, APLP1 and APLP2. Further, APP knockout mice are viable and minimally abnormal, while APP/APLP2 and APP/APLP1/APLP2 triple knockouts show perinatal lethality, making evaluation of the role of APP family members to brain aging and age-related neurodegenerative disease challenging[3]. Analysis of APP function in *Drosophila* is an attractive alternative[4]. Flies have one APP family member, Appl, and homozygous null flies are viable[5]. We recovered Appl in a forward genetic screen designed to uncover proteins and pathways

[1]Department of Pathology, Brigham and Women's Hospital, Harvard Medical School, Boston, Massachusetts, USA. [2]Aligning Science Across Parkinson's (ASAP) Collaborative Research Network, Chevy Chase, MD 20815, USA. [3]Department of Biological Engineering, Massachusetts Institute of Technology, Cambridge, MA, USA. [4]MIT Ph.D. Program in Computational and Systems Biology, Cambridge, MA, USA. [5]Genomics and Bioinformatics Hub, Brigham and Women's Hospital, Boston, MA, USA. ✉e-mail: mel_feany@hms.harvard.edu

required during aging to maintain neuronal viability. Given the importance of Appl in age-related cognition and neurodegenerative disease we performed a comprehensive transcriptomic and proteomic analysis of Appl mutant flies, integrated our data using network approaches and then validated and mechanistically analyzed a lead candidate pathway, control of proteostasis, using fly and mouse genetics and studies in human neurons (Fig. 1a).

## Results

### Neurodegeneration in *Appl* mutant flies

Identification of genes, like APP, responsible for genetic forms of neurodegenerative diseases in human patients has provided important insights into the mechanisms require to maintain neuronal function and viability with age. We took an alternative and complementary approach to the same problem in the facile genetic model organism

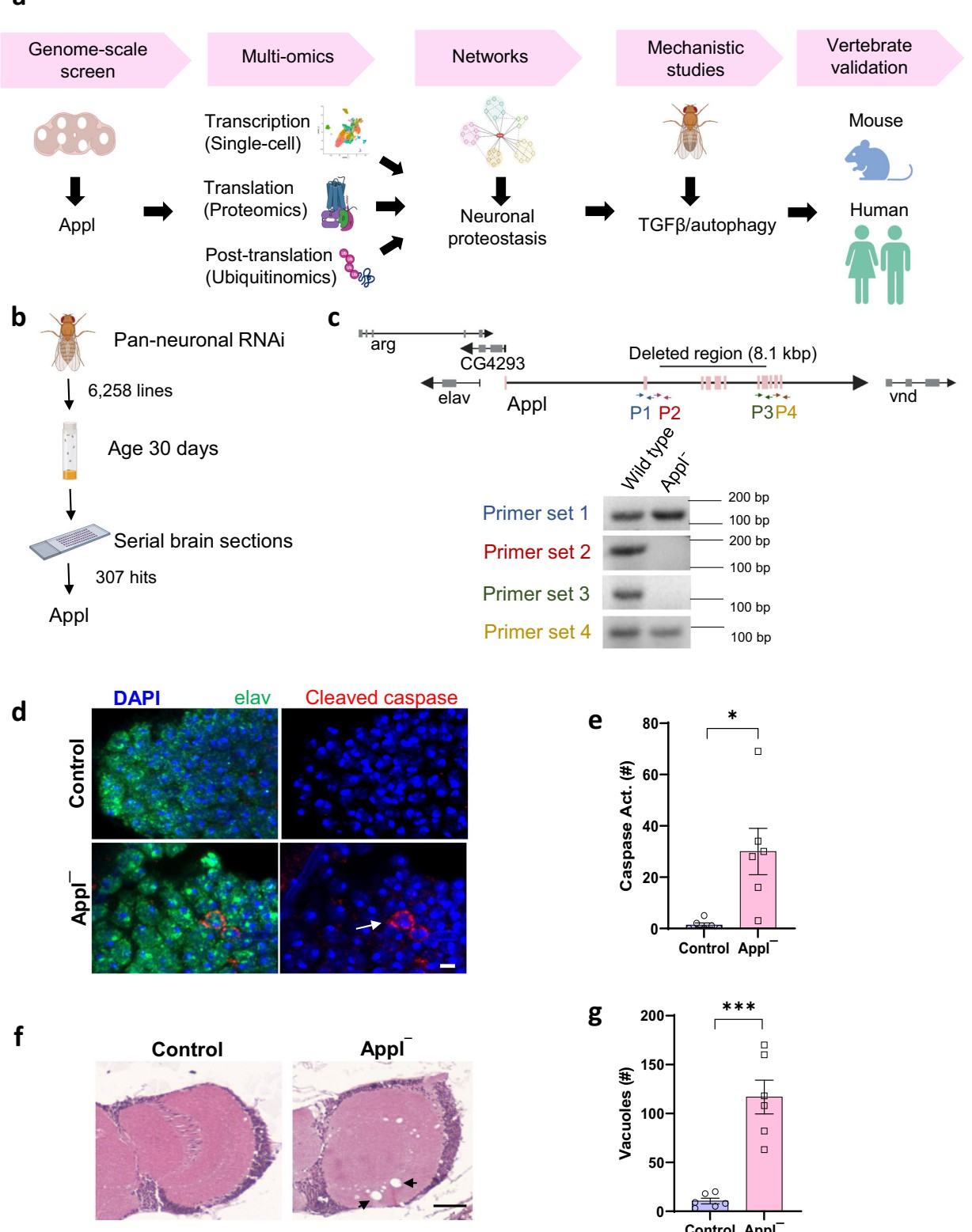

**Fig. 1 | Neurodegeneration in _Drosophila_ with loss of Appl. a** Schematic of study design to understand the function of Appl in the aging nervous system. **b** Unbiased transgenic RNAi screen identifies Appl as controlling neuronal viability with age. **c** The genomic region containing _Appl_ and adjacent transcription units. The indicated primers were used to define sequences retained and deleted in _Appl$^d$_ mutants, as indicated by DNA gels. F indicates forward primer, and R indicates reverse primer. Genotyping was performed using 5 biologically independent samples to confirm the loss of Appl with multiple primer pairs. Length of the polymerase chain reaction (PCR) products are indicated as base pairs (bp). **d** Representative immunofluorescence images with staining for elav (green), DAPI (blue), and transgenic caspase reporter, PARP (arrow, red). **e** Quantification shows

increased number of neurons with caspase activation in the whole brains of _Appl$^-$_ mutants compared to controls. p (Control vs _Appl$^-$_) = 0.0249). **f** Representative images of hematoxylin and eosin-stained brain sections showing increased numbers of vacuoles in _Appl$^-$_ mutants compared to controls. Arrows indicate vacuoles. **g** Quantitative analysis shows increased numbers of vacuoles (arrow) in the whole brains of _Appl$^-$_ flies compared to controls. p (Control vs _Appl$^-$_) = 0.0014). Control is _UAS-CD8-PARP-Venus, nSyb-GAL4/+_ in (**d**, **e**) and _nSyb-GAL4/+_ in (**f**, **g**). *$p < 0.05$, **$p < 0.01$, ***$p < 0.001$, two-tailed Student's t-test (**e**, **g**). Data are represented as mean ± SEM. $n = 6$ per genotype. Scale bars are 2 μm in (**d**) and 50 μm in (**f**). Flies are 20 days old. Created with BioRender.com (**a**, **b**). Source data are provided as a Source Data file.

_Drosophila_. We performed an unbiased forward genetic screen in which transgenic RNAi was used to knock down individual genes in a pan-neuronal pattern in otherwise normal animals. Adult flies were then aged to 30 days. Flies live for approximately 60 days under our culture conditions. Sections were taken through whole fly heads and the brain examined for histologic evidence of neurodegeneration (Fig. 1b). We assessed 6,258 lines targeting genes conserved from _Drosophila_ to humans and recovered 307 genes required for neuronal viability with age (Leventhal, Fraenkel and Feany, in preparation). One of the RNAi lines recovered targeted _Appl_, the gene encoding the single fly ortholog of mammalian APP family members.

Since RNAi can have off target effects, we verified _Appl_ as an authentic hit in our screen by obtaining a mutation in the _Appl_ gene, _Appl$^d$_. The _Appl$^d$_ allele was created as a synthetic deletion removing the central portion of the _Appl_ gene[5]. We used PCR and primers flanking and within the deletion to confirm that the majority of the _Appl_ coding region is deleted in _Appl$^d$_, while surrounding transcription units are not included in the deficiency (Fig. 1c). These observations are consistent with documented lack of Appl protein expression in _Appl$^d$_ mutants[6–9]. We next confirmed neurodegeneration in flies lacking Appl. We began by using a well-characterized transgenic reporter of caspase activity, which correlates well with cell death in _Drosophila_ models of neurodegeneration[10–12]. These reporter flies carry a transgenic construct in which the extracellular and transmembrane domains of mouse CD8 is fused to 40 amino acids from human PARP, including the caspase cleavage site of PARP (_UAS-CD8-PARP-Venus_)[13]. Endogenous _Drosophila_ caspases cleave the reporter at the PARP cleavage site. Caspase activation is then assessed experimentally using an antibody that is specific for cleaved human PARP. We found increased caspase activation in neurons of aged flies lacking Appl (Fig. 1d, arrow, e). We also observed increased numbers of neurodegenerative vacuoles in the neuropil and cortex of brains without Appl expression (Fig. 1f, arrows, g). Vacuole formation frequently accompanies neurodegeneration in _Drosophila_[14–17] and has previously been observed in flies lacking Appl function[8].

**Multiomic analysis identifies diverse cellular pathways perturbed by loss of Appl**
We began to probe the proteins and pathways underlying the neurodegeneration observed in flies lacking Appl function by performing single-cell RNA sequencing (scRNA) to identify transcripts regulated by Appl in different cell types. Using an optimized brain dissociation method, 10x library preparation, sequencing, and a bioinformatics analysis pipeline, we implemented scRNA sequencing on _Appl$^d$_ (_Appl$^-$_) mutant and control _Drosophila_ brains (Fig. 2a). Following sequencing and quality control, cells were clustered and annotated based on the enrichment of marker genes in the cell population (Supplementary Fig. 1a). We identified 24 unique cell clusters. Each cluster was present in both control and _Appl$^-$_ brains and 17 of the clusters were neuronal (Fig. 2b). We also identified non-neuronal clusters, including glia, hemocytes, fat body, cone cells, a mix of hemocytes and mushroom body output neurons, and an unannotated cluster containing markers for multiple types neurons (Fig. 2b).

We next performed differential gene expression analysis to identify genes modulated by removing Appl function. We observed 720 upregulated and 717 downregulated genes with 1.25-fold or greater change in gene expression and 0.05 adjusted _p_-value. Up and downregulated genes were present in each cell cluster (Supplementary Fig. 1b). Transcripts up and downregulated were then ranked by the number of clusters in which they showed altered expression. In the upregulated group of genes, there was notable enrichment for mitochondrial genes (Fig. 2c, Supplementary Fig. 1c). In contrast, there was pervasive downregulation of genes involved in protein synthesis, including translation elongation and structural constituents of ribosomes (Fig. 2c, Supplementary Fig. 1c).

Gene ontology analysis of all differentially expressed genes shows that Appl regulates transcript levels of genes involved in translation, axon and dendrite development and function, synaptic function, cell adhesion, and long-term memory, among others (Fig. 2d). Of these, translation-related genes were significantly enriched in gene ontology analysis with the highest combined score. Since Appl appears to be expressed primarily in neurons[18], and many of the GO terms in Fig. 2d highlighted neuronal functions, we examined neuron-specific gene expression in more detail. We analyzed downregulated and upregulated genes separately using GO analysis. Many of the same enriched pathways were found upon examination of neuron-specific transcriptional changes (Supplementary Fig. 2a). In addition, TGFß/BMP signaling was identified in the downregulated neuronal gene set. Focusing specifically on signaling pathways identified by GO analysis we observed significant changes in a number of other cellular signaling systems as well upon loss of Appl function (Table 1).

Although Appl is expressed primarily in neurons, we also identified downregulated and upregulated genes in non-neuronal cells, including in intrinsic brain cells (glia) and attached tissue (fat body) (Supplementary Fig. 1b, Supplementary Fig. 2b). The transcriptional effects of removing Appl function throughout the animal on different cell types are largely distinct. Most neuronal subtypes have similar sets of altered genes and fewer unique genes (Supplementary Fig. 2b). In contrast, the number of unique genes is higher in glial cells and fat body (Supplementary Fig. 2b). For both glia (Supplementary Fig. 2c) and fat body (Supplementary Fig. 2d), the majority of differentially expressed genes are unique. GO analysis supports different transcriptional profiles in distinct cell types in response to loss of Appl at the pathway level, with little overlap among the most significantly altered molecular functions in neurons, glia and fat body (Supplementary Fig. 2c, d).

Next we performed proteomics and ubiquitinomics on whole fly heads using mass-spectrometry to identify the proteins translationally and post-translationally regulated by Appl. We used antibodies that recognize diglycine-modified lysine residues present at ubiquitin-modified sites to identify ubiquitinated proteins (Fig. 3a). We identified 6364 proteins in the whole proteome analysis. Of these, 545 proteins showed 1.25-fold or greater upregulation and 679 showed downregulation in protein levels with 0.05 FDR-adjusted _p_-value (Fig. 3b). In our ubiquitinomics analysis, we identified 2100 ubiquitinated sites representing 1069 proteins. Of these, 41 proteins

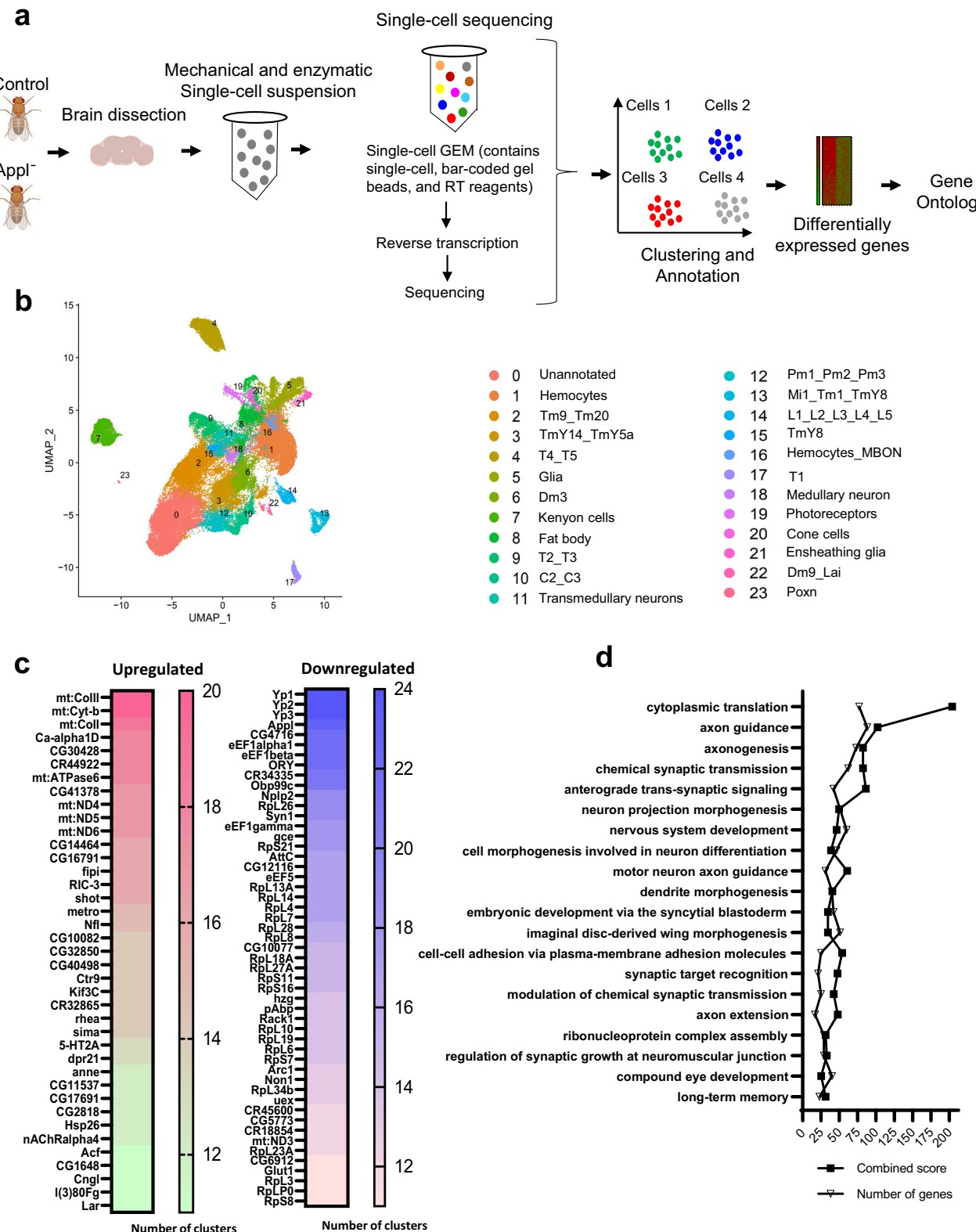

**Fig. 2 | Single-cell RNA-sequencing in _Drosophila_ brains with loss of Appl.**
**a** Schematic of study design for single-cell RNA sequencing (scRNA-seq). Created with BioRender.com. **b** UMAP plot of scRNA-seq data from the fly brain shows clusters of neuronal and non-neuronal cell populations. Each color represents a distinct cell cluster as indicated. **c** Genes upregulated and downregulated in more than 10 cell clusters are shown. The given numbers represent the number of clusters with differential expression of the corresponding gene. **d** Gene ontology analysis of differentially expressed genes using FlyEnrichr categorizes genes under the biological process. The _p_-value is computed using Fisher's exact test, a binomial proportion test assuming a binomial distribution. The combined score was calculated by combining the z-score and _p_-value using the formula $c = \ln(p) * z$. Source data are provided as a Source Data file.

**Table 1 | Signaling pathways affected by loss of Appl**

| Gene ontology term | p-value | z-score | Combined Score | Genes |
|---|---|---|---|---|
| TGFβ/BMP signaling | 1.39E-06 | −675.7 | 9113.1 | babo;kn;mei-P26;nkd;cv-c;Cip4;Vap33;put;nmo;Actbeta;scrib;bsk;Rac1;Wnt4;eIF4A |
| Ras protein signal transduction | 0.0001 | −535.8 | 4743 | hid;jeb;S6k;RyR;Octbeta3R;aPKC;Graf;crb;RhoGEF2;scrib;Eip93F;bsk;cic |
| Hippo signaling | 0.0008 | −546.7 | 3890.4 | crb;sas;cbt;rin;Actbeta;Nedd4;scrib;aPKC;bsk;cic |
| ERBB signaling | 0.003 | −543.7 | 3185.5 | Graf;sas;Syx1A;Tl;Eip93F;emc;Vav;bsk;Rac1;dock;cic |
| Wnt signaling | 0.008 | −469.7 | 2255.8 | Frl;bsk;Ssdp;Galphao;Wnt4 |
| Glutamate receptor signaling | 0.0046 | −411.4 | 2209.1 | Syt4;SPR;Actbeta;cv-c;CASK;mGluR |
| TORC1 signaling | 0.001 | −320.6 | 2056.6 | rin;Eip93F;bsk;Lpin;eIF4A |
| Notch signaling | 0.0001 | −190.0 | 1650.4 | Rbfox1;ct;crb;pros;heph;Nedd4;scrib;mam;emc;aPKC;bsk |
| G-protein coupled glutamate receptor signaling | 0.012 | −372.6 | 1628.2 | SPR;mGluR |
| calcium-mediated signaling | 0.02 | −253.6 | 988.5 | Fife;Piezo;RyR |

Signaling pathways identified through the gene ontology analysis of neuron-specific genes altered in Appl mutant flies categorized under biological process. Gene ontology was performed using FlyEnrichr. Gene ontology terms identified as signaling pathways with p-value < 0.05 are listed. The combined score is calculated by multiplying the p-value and z-score: c = ln(p) * z. The p-value is computed using Fisher's exact test, a binomial proportion test assuming a binomial distribution.

showed 1.25-fold or greater upregulation of ubiquitination and 128 showed downregulation in protein levels with 0.1 FDR adjusted p-value (Fig. 3c). In both proteomic and ubiquitinomic data we observed a preponderance of downregulated proteins, including many proteins involved in translation (Supplementary Data 1), consistent with downregulation of translation at the transcriptional level (Fig. 2c, d).

We then used the solution of the prize-collecting Steiner forest (PCSF) algorithm[19] to map transcriptomic, proteomic and ubiquitinomic data onto a network of physical protein interactions using human interactome data. We found multiple subnetworks involved in cellular pathways not previously implicated in APP biology, including translation, mitochondrial function, nucleic acid and lipid metabolism, and proteostasis (Fig. 3d). Interestingly, three processes key for maintaining cellular protein homeostasis, or proteostasis[20], including protein synthesis, protein folding, and protein degradation, were altered following removal of Appl. Since protein aggregation has been strongly implicated in the pathogenesis of Alzheimer's disease and related disorders, but not previously linked to endogenous Appl function, we focused on validation and mechanistic exploration of the control of protein homeostasis by APP.

**Appl regulates proteostasis via TGFβ signaling and autophagy**
The number of ubiquitin-positive protein aggregates increases with age in Drosophila and mammalian brains, and is further increased with compromise of proteostasis mechanisms[21,22]. We therefore began our investigation of the role of Appl in regulating proteostasis by immunostaining with an antibody directed to ubiquitin. We observed an increase in the number of ubiquitin-positive aggregates in the retinas and brains of flies lacking Appl expression. Aggregates were particularly prominent in the retina (Fig. 4a, arrows), a site of documented Appl expression[23] with well-defined anatomy amenable to quantitative analysis. The numbers of ubiquitinated aggregates increased with age in both control flies and flies without Appl, with earlier and greater accumulation in mutant flies (Fig. 4b), confirming an aging-dependent proteostasis defect due to the loss of Appl. Aggregates were stained with the commonly used ProteoStat dye (Supplementary Fig. 3a, b), which binds specifically to aggregated protein. As expected[21,24,25], aggregate formation was accompanied by increased levels of biochemically insoluble ubiquitinated protein (Supplementary Fig. 3c, d). Immunostaining with antibodies directed to the early endosomal small GTPase Rab5 and the recycling endosome marker Rab11 failed to show colocalization with ubiquitinated aggregates (Supplementary Fig. 3e, f), suggesting that ubiquitin-positive aggregates do not represent abnormal endosomes.

We additionally verified the role of Appl in controlling neuronal proteostasis by overexpressing Appl in an attempt to rescue defective proteostasis. We observed significantly reduced numbers of ubiquitin-positive aggregates in Appl⁻ flies overexpressing Drosophila Appl in neurons using the pan-neuronal GAL4 driver nSyb-GAL4 (Fig. 4c, arrows, d). We next asked if human APP could substitute functionally for fly Appl. When we overexpressed the 695 amino acid isoform of human APP in neurons we observed significantly reduced numbers of aggregates in Appl⁻ flies (Fig. 4c, d), consistent with a conserved role for APP family members in controlling proteostasis.

We then turned to the mechanism by which Appl regulates proteostasis. Appl has previously been implicated in intercellular signaling[3] and our transcriptional analysis (Table 1) indeed identified a number of candidate signaling pathways influenced by loss of Appl function. Since signaling pathways are enriched for proteins, such as transmembrane receptors and kinases, which are traditionally good therapeutic targets, and the control of proteostasis by cellular signaling cascades is not well understood, we focused on cellular signaling. We obtained transgenic RNAi lines targeting key members of the top signaling pathways identified in our transcriptional studies, TGFß/BMP. Genes encoding critical regulators of the TGFß/BMP signaling pathway were knocked down in a pan-neuronal pattern in the presence and absence of Appl and the numbers of ubiquitin-positive aggregates in the retina and brain were assessed by immunostaining. When we used transgenic RNAi to reduce TGFβ signaling in neurons by knocking down the TGFβ ligand Activin-β, receptor subunit punt, or major transcription factor Smad on X (Smox) (Fig. 5a), we observed increased numbers of ubiquitin-immunoreactive retinal aggregates flies lacking Appl (Fig. 5b, c). We then assessed the activity of the TGFβ signaling pathway by measuring the levels of activated, phosphorylated TGFβ transcription factor Smox. We observed decreased Smox phosphorylation in Appl⁻ flies (Fig. 5d). Similarly, we found decreased levels of a well-documented transcriptional target of TGFβ signaling, EcRB1, in flies lacking Appl (Fig. 5e). These findings together suggest that Appl function is required to maintain TGFβ in the adult retina, and further implicate Appl-mediated TGFβ in control of proteostasis during aging.

We next determined how TGFβ regulated protein aggregation in flies lacking Appl. Since activin signaling has previously been found to regulate macroautophagy[26,27], henceforth autophagy, we hypothesized that the proteostasis defect observed in flies lacking Appl function could be due to defective autophagy. The microtubule-associated protein light chain 3 (LC3) is cleaved, processed, and inserted into nascent autophagosomes, where it is involved in both autophagosome formation and selection of targets for degradation[28]. The Autophagy-

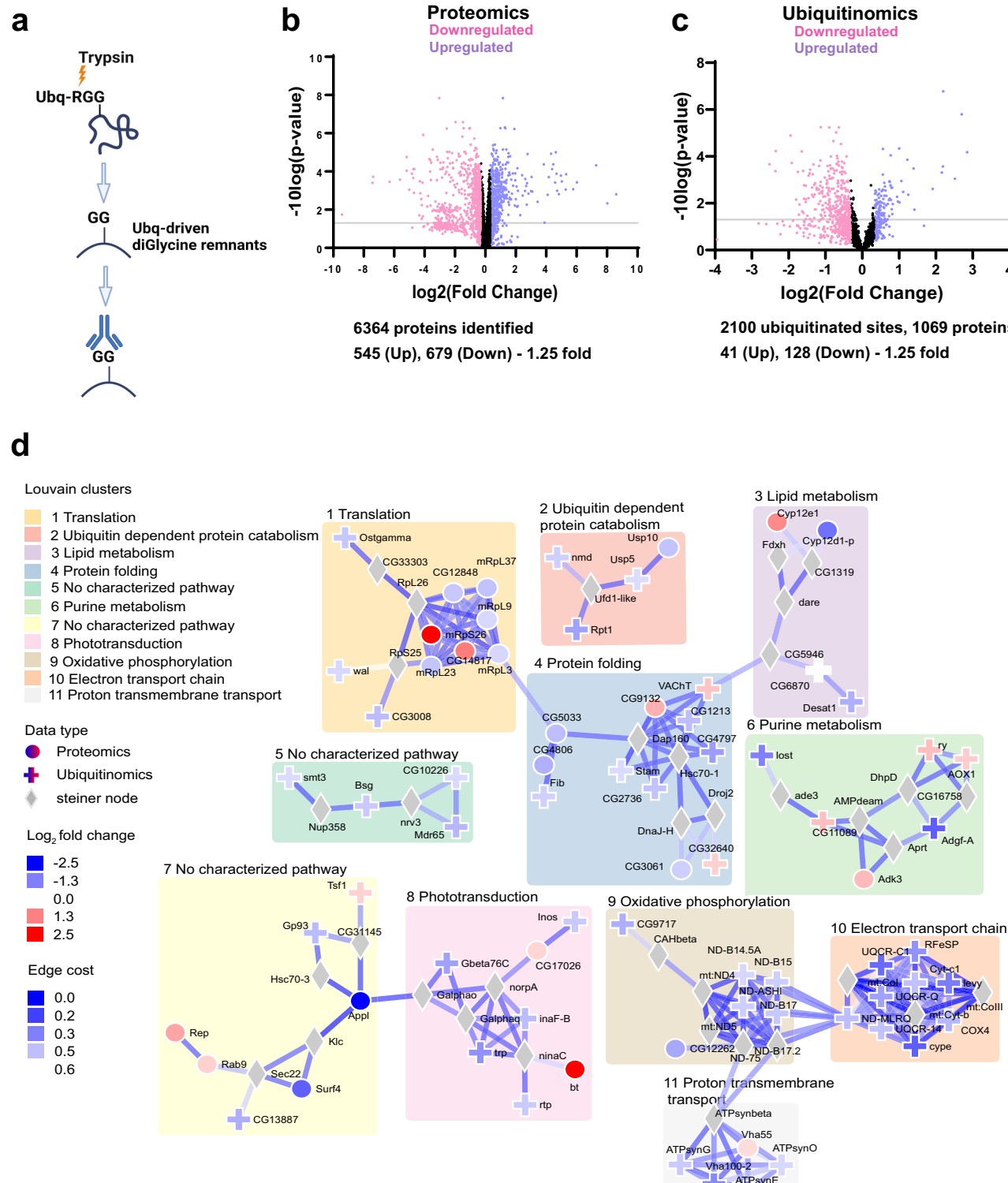

**b** Proteomics

6364 proteins identified
545 (Up), 679 (Down) - 1.25 fold

**c** Ubiquitinomics

2100 ubiquitinated sites, 1069 proteins
41 (Up), 128 (Down) - 1.25 fold

related gene 8a (Atg8a) protein is the fly homolog of human LC3 and is widely used to mark autophagic structures in *Drosophila*. We identified autophagosomes using an antibody recognizing fly Atg8a[28–30]. Immunostaining revealed increased numbers of Atg8a-immunoreactive puncta in *Appl*[-] flies compared to controls (Fig. 6a, arrow, b).

The p62 protein, also known as sequestosome 1 (SQSTM1), promotes degradation of ubiquitinated proteins and protein aggregates by directing ubiquitin-modified proteins to LC3. Upon autophagic activation, p62 is recruited to autophagosomes and eventually degraded in lysosomes. p62 accumulation frequently accompanies

aberrant autophagy and correlates with reduced autophagic flux[31,32]. The *Drosophila* homolog of p62, ref(2)P, is a component of protein aggregates formed in brain and peripheral tissues under conditions of disrupted autophagy, including neurodegenerative disease and aging[33]. Immunofluorescence with an antibody directed to *Drosophila* p62[34] showed increased numbers of p62-positive puncta in the retina of *Appl*[-] flies compared to controls (Supplementary Fig. 4a, arrow, b). As expected[33,35], occasional ubiquitin-positive aggregates co-stained for Atg8a or p62 (Supplementary Fig. 4c, d, arrows). We used the well-validated tandem GFP-mCherry-Atg8a reporter[31,36] to confirm impaired

**Fig. 3 | Proteome, ubiquitinome and integrative analysis of flies with loss of Appl. a** Schematic of enrichment procedure for ubiquitinated proteins using the diglycine antibody, which targets the GG remnants of ubiquitinated regions following trypsin digestion. Created with BioRender.com. **b** Volcano plot showing the distribution of proteins in the whole proteome analysis of Appl deleted flies. Protein level changes greater than 1.25-fold are indicated as pink and violet dots. Violet dots indicate upregulated proteins; pink dots indicate down-regulated proteins. The grey line marks False Discovery Rate (FDR) adjusted two-tailed *p*-value = 0.05. Proteins indicated by dots below that line are not considered statistically significant. **c** Volcano plot showing the distribution of ubiquitinated proteins in Appl deleted flies. DiGlycine peptide changes, indicating ubiquitination, above 1.25-fold, are indicated as pink and violet dots. Violet dots indicate upregulated proteins; pink dots indicate down-regulated proteins. The grey line marks False Discovery Rate

(FDR) adjusted two-tailed *p*-value = 0.05. Proteins indicated by dots below that line are not considered statistically significant. **d** Protein networks altered by removing Appl. Louvain clustering of proteins altered in proteome and ubiquitinome analysis of flies without Appl show enrichment of proteostasis-related networks such as translation, protein folding, and ubiquitin-dependent degradation. Shapes of the nodes show the source of protein. Steiner nodes are the proteins that are not experimentally altered; however, they are predicted to be associated with loss of Appl. Results from single-cell RNA sequencing data were used to adjust edge confidence and node weights in the network. The cost of the edges is determined by the confidence in the protein-protein interaction as calculated by STRING. These edges were weighted by the number of single cell clusters with altered candidate gene expression. A lower cost indicates greater confidence in the protein-protein interaction. Source data are provided as a Source Data file.

autophagic flux in flies lacking Appl function (Supplementary Fig. 4e, f).

As for ubiquitinated aggregates, we assessed the role of TGFβ signaling on markers of autophagy in flies lacking Appl function and controls. We reduced the levels of the TGFβ ligand Activin-ß, receptor subunit punt, or transcription factor Smox in neurons in *Appl⁻* and control flies. We found that each of the transgenic RNAi lines further increased the number of Atg8a-positive (Fig. 6c) or p62-positive (Supplementary Fig. 5) retinal puncta in flies lacking Appl function, consistent with our prior findings when monitoring the number of ubiquitin-immunoreactive inclusions (Fig. 5c).

Given the apparent importance of APP processing in the pathogenesis of Alzheimer's disease, we next assessed the ability of a transgenic construct encoding the secreted extracellular domain of Appl[37] (Supplementary Fig. 6a) to influence neuronal proteostasis. Neuronal expression of secreted Appl (Appl-s) was able to rescue the increase in ubiquitin-immunoreactive (Supplementary Fig. 6b) and Atg8a-positive (Supplementary Fig. 6c) puncta present in *Appl⁻* retinas. Thus, the extracellular sequences of Appl contain the residues critical for protection from loss of proteostasis observed in the absence of Appl function.

Smox has previously been shown to repress *Atg8a* expression in muscle in *Drosophila* and to bind directly to the *Atg8a* promoter[26]. We used the sensitive RNA in situ hybridization assay RNAScope to examine transcription of *Atg8a* in control and *Appl⁻* retinas (Fig. 6d). Quantitative analysis revealed significantly increased levels of *Atg8a* RNA in the absence of Appl (Fig. 6e), consistent with prior negative transcriptional regulation of *Atg8a* by Smox observed in muscle[26]. Consistent with the observed transcriptional upregulation, we observed increased Atg8a protein levels in *Appl⁻* flies (Supplementary Fig. 6d).

We next explored the functional effects of increasing Atg8a levels in the retina using an *Atg8a* transgene directed to retinal neurons. We found that increasing Atg8a levels resulted in formation of increased numbers of ubiquitinated proteins aggregates in the retina (Fig. 6f, arrow, g). The increase in ubiquitin-positive retinal aggregates seen in the Atg8a overexpressing animals (Fig. 6f, g) is consistent with upregulation of *Atg8a* seen with loss of Appl function (Fig. 6d, e, Supplementary Fig. 6d). Overexpressed Atg8a-GFP did not mislocalize to endosomes as assessed by lack of colocalization with endosomal markers (Supplementary Fig. 6e, f).

## APP deficiency impairs proteostasis and reduces TGFβ signaling in mice and in human neurons

Our data in *Drosophila* suggest that Appl plays an important role in maintaining proteostasis during aging by sustaining TGFβ regulation of autophagy. We began to investigate the applicability of our simple genetic model finding to vertebrates by examining brains of mice with genetic manipulation of APP family members. APLP2 is a close homolog and has an expression pattern similar to APP[3]. We examined mice in which both APP and APLP2 were deleted in neurons[38] to study

the function of APP without the compensatory effect of APLP2. Double knockout mice were created by first crossing mice with a floxed *APP* allele with transgenic mice expressing Cre-recombinase under the neuronal rat nest promoter (Nestin-Cre)[39]. These neuronal conditional knockout mice were then crossed onto an *APLP2* knockout background to create neuronal double conditional knockout mice (N-dCKO) (Fig. 7a). N-dCKO mice are viable and display defects in neuromuscular synapse patterning[38]. Sections were prepared from brains from 18-month-old N-dCKO and controls and immunostained with an antibody to ubiquitin. *APLP2* null animals were used as controls because these mice appear behaviorally, physiologically and neuropathologically normal[40–42]. We observed increased ubiquitin immunostaining in cortical neurons in N-dCKO mice compared to controls (Fig. 7b, c). Similarly, N-dCKO neurons contained increased levels of LC3B (Fig. 7d, e) and of the related Atg8 family member GABARAP (Supplementary Fig. 7a, b). Next, we assessed these brains for evidence of altered TGFβ signaling. Phosphorylation of SMAD3, one of the five receptor-activated mammalian SMADs[43], was reduced in N-dCKO mice compared to controls (Fig. 7f, g).

We next addressed the function of APP in human neurons using induced pluripotent stem cell (iPSC)-derived APP knockout neurons and isogenic controls (Fig. 8a, Supplementary Fig. 8a). We expressed the neuronal fate-inducing transcription factor Neurogenin2 (NGN2) to differentiate iPSCs into neurons (Fig. 8a, Supplementary Fig. 8a)[44]. GFP expression is induced together with NGN2 and allows visualization of neuronal morphology (Supplementary Fig. 8a). We verified loss of APP in knockout neurons by western blotting (Supplementary Fig. 8b). Previous studies have shown that knockout iPSC neurons do not have altered APLP1 or APLP2 levels[45]. We examined the effect of APP loss on proteostasis, autophagy markers, and TGFβ activity in human neurons. As observed in knockout fly and mouse brains, loss of APP led to elevated levels of cytoplasmic ubiquitin in knockout neurons compared to controls (Fig. 8b, c). We also observed increased numbers of puncta immunopositive for the autophagy markers LC3B (Fig. 8d, arrows, e) or GABARAP (Supplementary Fig. 9a, arrow, b) in APP knockout neurons compared to controls. We used western blotting analysis to monitor conversion of GABARAP-I to GABARAP-II. GABARAP-I is converted to GABARAP-II by conjugation to phosphatidylethanolamine, which mediates autophagosome membrane attachment[46]. APP knockout neurons had an increase in the ratio of GABARAP-I processed to GABARAP-II (Supplementary Fig. 9c, d), consistent with altered autophagy.

We then assessed TGFβ signaling by immunostaining and western blotting with an antibody recognizing phosphorylated SMAD3. We observed decreased nuclear phospho-SMAD3 in APP knockout neurons compared to controls by quantitative immunofluorescence (Fig. 8f, g) and by western blotting (Supplementary Fig. 9e, f). We also employed an antibody recognizing the phosphorylated form of additional receptor-activated mammalian SMADs, SMAD1/5/9, but did not observe consistently altered phospho-SMAD levels in APP knockout neurons. Consistent with our in vivo results in flies and mice, APP

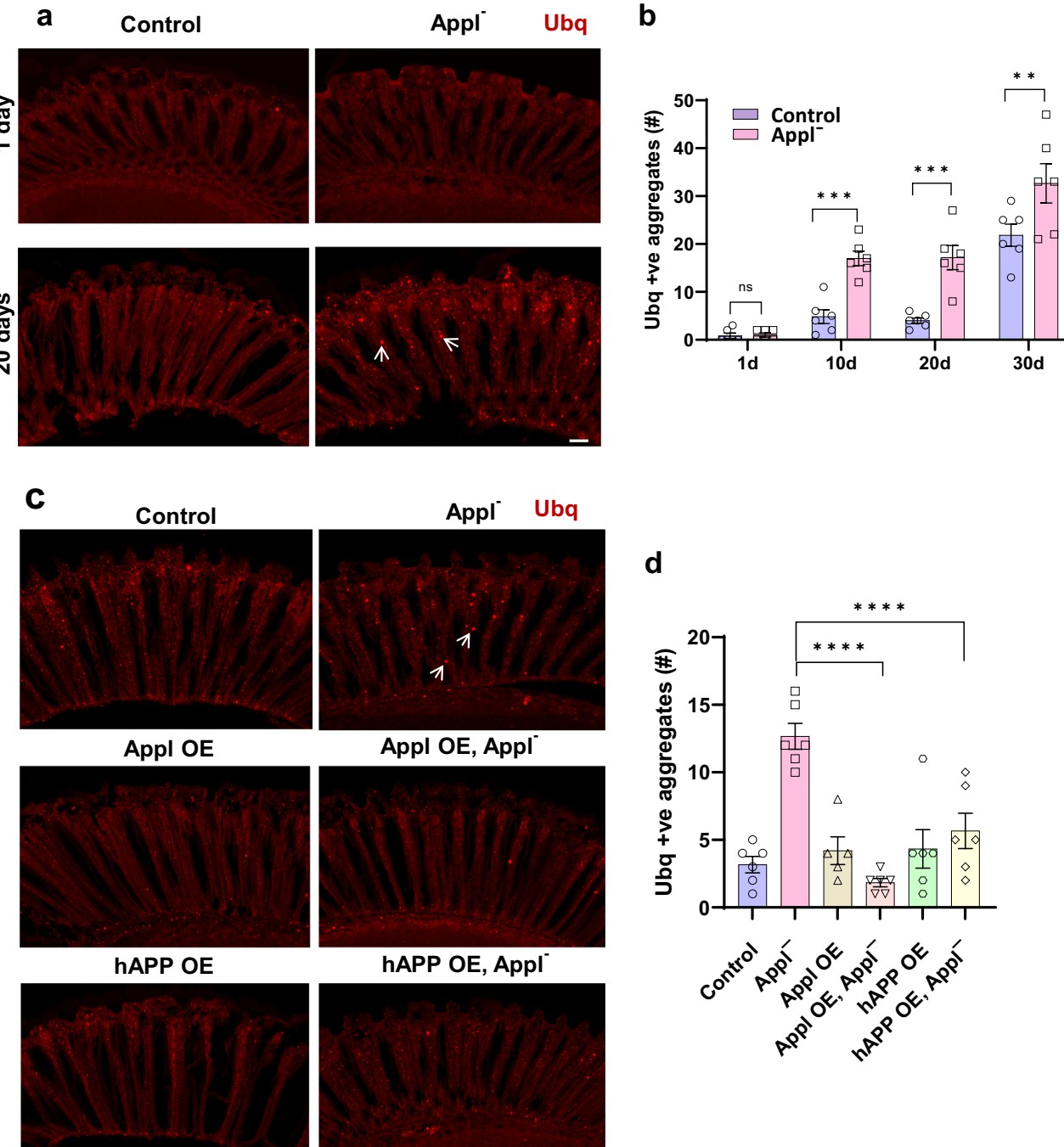

**Fig. 4 | Appl loss leads to ubiquitinated protein aggregate formation.**
**a** Representative immunofluorescent staining of ubiquitin-positive aggregates (arrows) in 1-day-old and 20-day-old fly retinas. **b** Quantification shows the number of ubiquitin-positive aggregates in retinal sections in 1-, 10-, 20- and 30-day-old *Appl⁻* flies compared to controls. *p*-values for Control 10d vs *Appl⁻* 10d, Control 20d vs *Appl⁻* 20d, Control 30d vs *Appl⁻* 30d are 0.0002, 0.0002, and 0.0021 respectively. **c** Representative images illustrate reduction of ubiquitin-positive aggregates (arrows) in retinas of *Appl⁻* flies overexpressing fly Appl or human APP.

**d** Quantification demonstrates significant reduction of retinal ubiquitinated aggregates by increasing expression of Appl or human APP in retinal neurons. *p*-value for *Appl⁻* vs Appl OE, *Appl⁻* is <0.0001 and *Appl⁻* vs hAPP OE, *Appl⁻* is <0.0001. Control is *nSyb-GAL4/+*. **p < 0.01, ***p < 0.001, ****p < 0.0001. Two-way (**b**) and one-way (**d**) ANOVA with Student-Newman-Keuls posthoc test. Data are represented as mean ± SEM. *n* = 6 per genotype. Scale bars are 10 μm. Flies are the indicated age in (**b**) and 10 days old in (**d**). Source data are provided as a Source Data file.

knockout human neurons showed altered proteostasis and reduced TGFβ signaling.

## Loss of Appl promotes tauopathy in vivo
Our findings thus far identify a new role for APP in controlling age-dependent proteostasis, but do not directly address the relevance to disease. The diagnosis of Alzheimer's disease at postmortem

examination requires not only the presence of Aβ plaques, but also intracellular neurofibrillary tangles comprised of aggregated wild type tau, a microtubule binding protein. We have previously described a *Drosophila* model for the study of Alzheimer's disease and related tauopathies based on the pan-neuronal expression of wild type or familial frontotemporal dementia-linked mutant forms of human tau[15]. We examined the influence of loss of Appl on tau neurotoxicity by

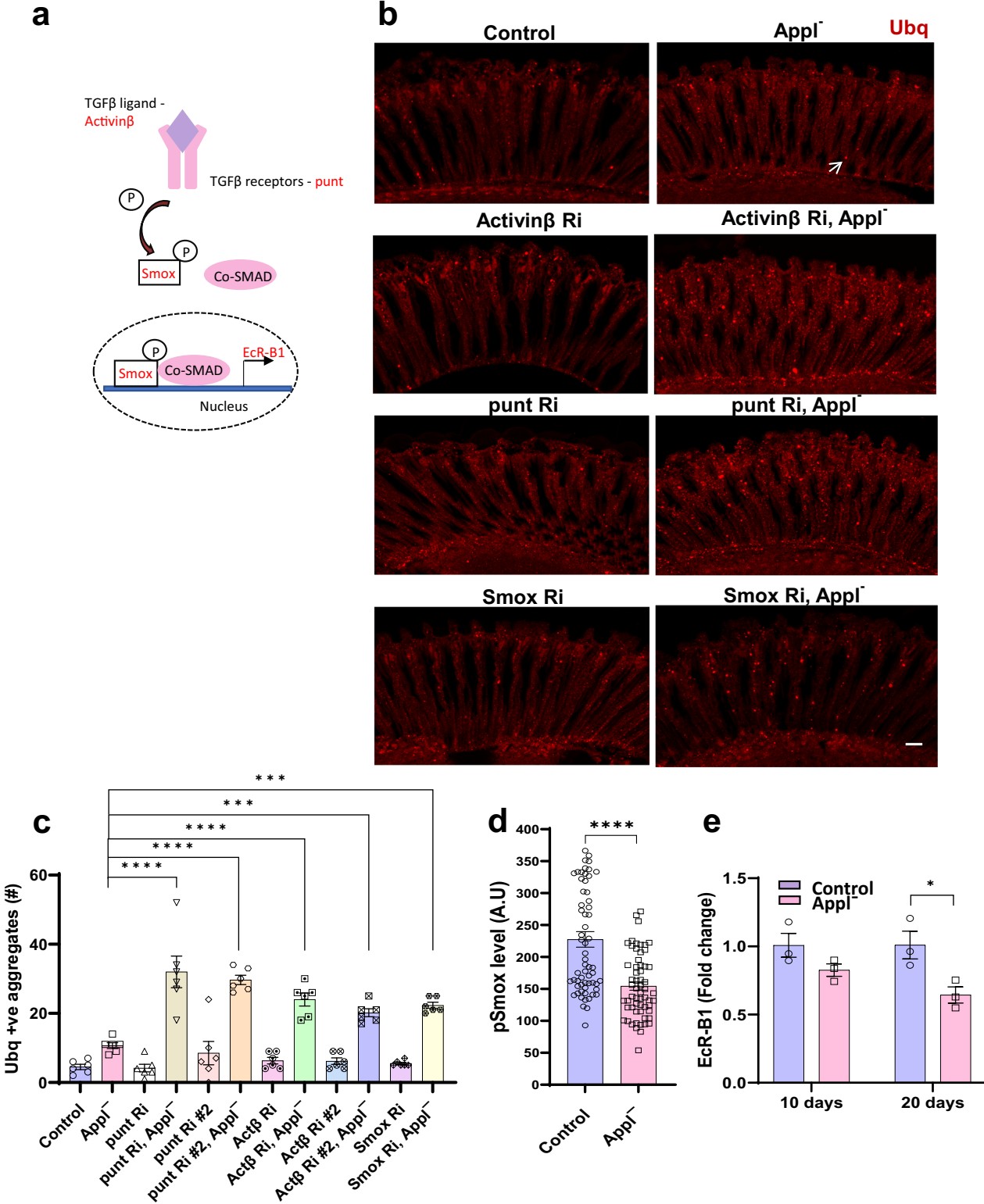

creating Appl deleted flies expressing transgenic human tau (Fig. 9a). Expression of wild type human tau in a pan-neuronal pattern produced neuronal dysfunction and death as assessed with locomotor activity monitored by the climbing assay (Fig. 9b). Removing Appl function in human tau transgenic flies further impaired locomotor function (Fig. 9b).

We assessed neurodegeneration in human tau transgenic flies with and without expression of Appl by monitoring caspase activation using a transgenic caspase reporter and by quantifying vacuolar pathology. In both the caspase activation and vacuole assays we found that removing Appl significantly enhanced tau-mediated neurodegeneration (Fig. 9c, d). We also saw exacerbation of lifespan truncation in flies lacking Appl and expressing human tau (Supplementary Fig. 10a). Tau neurotoxicity is strongly dependent on phosphorylation[47–50]. We did not find clear alteration of well-studied tau phosphoepitopes in flies expressing human tau in the absence of Appl using whole head homogenates (Supplementary Fig. 10b), suggesting that Appl may work downstream of tau phosphorylation.

**Fig. 5 | Appl regulates proteostasis through the TGFβ signaling pathway.**
**a** Schematic of TGFβ signaling pathway components assessed. **b** Representative immunofluorescent staining showing ubiquitin-positive aggregates (arrow) in the retinas of *Appl⁻* flies expressing transgenic RNAi directed to the TGFβ ligand (*Activinβ*), TGFβ receptor (*punt*), and transcription factor (*Smox*) in retinal neurons. **c** Quantification of the number of ubiquitin-positive aggregates in retinal sections of control and *Appl⁻* flies with TGFβ pathway manipulation by transgenic RNAi in retinal neurons. *p*-value for ubiquitin-positive aggregates: *Appl⁻* vs punt Ri *Appl⁻*, < 0.0001; *Appl⁻* vs punt Ri #2 *Appl⁻*, <0.0001; *Appl⁻* vs Actβ Ri *Appl⁻*, <0.0001; *Appl⁻* vs Actβ Ri #2 *Appl⁻*, 0.0002; *Appl⁻* vs Smox Ri *Appl⁻*, 0.0002. **d** Quantitative

analysis of the fluorescent intensity of phosphorylated Smox (pSmox), a TGFβ signaling marker, shows reduction in *Appl⁻* flies compared to controls. *p*-value (Control vs *Appl⁻*) = 7.61E-07. **e** Real-time PCR quantifying *EcR-B1* transcript levels, a target gene of the TGFβ signaling pathway shows decreased transcripts in *Appl⁻* flies compared to controls. *p* (Control 20d vs *Appl⁻* 20d) = 0.046. Control is *nSyb-GAL4/+*. \**p* < 0.05, \*\**p* < 0.01, \*\*\**p* < 0.001, \*\*\*\**p* < 0.0001, one-way ANOVA with Student-Newman-Keuls posthoc test (c) and two-tailed Student's t-test (d, e). Data are represented as mean ± SEM. *n* = 6 per genotype (a), *n* = 60 cells per genotype (d), *n* = 3 per genotype (e). Scale bar is 10 μm. Flies are 10 days old in (**b**–**d**) and the indicated age in (**e**). Source data are provided as a Source Data file.

## Discussion

Here we take a multi-omic approach to identify previously unsuspected, pervasive control of cellular metabolism by Appl, the single *Drosophila* ortholog of the vertebrate APP family of proteins. Integrating single cell RNA sequencing with proteomics and ubiquitinomics from flies lacking Appl function and controls provides evidence for dysregulated translation, mitochondrial function, nucleic acid and lipid metabolism, and proteostasis in Appl mutants (Fig. 3d). Despite the dominant focus on Aβ peptides in Alzheimer's disease research, a number of functions for APP have nonetheless been defined using biochemical, cellular and genetic approaches[3,51]. Important roles for APP family members in cell adhesion[38,52], axon and synapse development and function[41,42,53,54], and neuronal excitability[55] have been described[3,51]. We identified many of these pathways in our analyses as well (Fig. 2, Supplementary Fig. 2). The molecular mechanisms underlying the previously described functions of APP are not fully understood and the new APP-regulated pathways we describe here are candidates for regulating these diverse roles of APP.

The strong age-dependence of Alzheimer's disease has been ascribed, at least in part, to loss of the normal mechanisms controlling protein synthesis, folding and degradation, or proteostasis, during aging[20,56]. Loss of normal proteostatic mechanisms can promote aggregation of protein into the histopathological hallmarks of Alzheimer's disease, the extracellular Aβ plaques and intracellular neurofibrillary tangles, as well as perturbing multiple other cellular pathways critical for function and viability[20]. We thus focused here on defining (Fig. 4) and mechanistically characterizing (Figs. 5, 6) a previously unsuspected role for APP in controlling proteostasis. Specifically, we found that TGBβ signaling, a top candidate from our transcriptional analysis (Supplementary Fig. 2, Table 1) controls age-dependent abnormal protein aggregation and autophagy in the *Drosophila* nervous system (Figs. 5, 6). We demonstrated the relevance of our findings to mammals by showing similar regulation in APP neuronal conditional knockout mice (Fig. 7) and in cultured human APP knockout neurons (Fig. 8). APP may regulate TGFβ signaling by binding to TGFβ ligand. Immunoprecipitation studies have shown that recombinant TGFβ2 can bind directly to the extracellular domain of APP in vitro[57]. We did not detect direct binding of either full length or secreted Appl to TGFβ ligands or receptor subunits in *Drosophila* brains in vivo, although our experiments were limited by available immunologic reagents. Thus, the precise mechanism of TGFβ regulation by APP requires further investigation. Nonetheless, we have shown that expressing the extracellular region of Appl can rescue loss of function phenotypes associated with loss of Appl function (Supplementary Fig. 6). Based on these results and prior reports showing a direct physical interaction between the extracellular domain of APP and TGFβ ligand[58,59], we hypothesize that Appl binds to the extracellular domain of TGFβ ligand, blocking receptor activation and thus reducing downstream phosphorylation of the Smox transcription factor (Fig. 5).

Our findings implicate autophagy, and specifically Atg8 (Fig. 6d, e), as the target of TFGβ signaling via the transcription factor SMAD. Smox represses *Atg8a* expression in *Drosophila* muscle and binds directly to the *Atg8a* promoter[26]. Our observation of increased *Atg8a*

RNA in the retinas of flies lacking Appl by in situ hybridization is consistent with prior results manipulating TFGβ signaling in muscle. Modest overexpression of Atg8a has previously been shown to reduce accumulation of ubiquitinated protein aggregates[24]. In contrast, we observed elevated numbers of protein aggregates when we increased Atg8a expression in adult fly retinal neurons either by removing Appl function or by overexpressing transgenic Atg8a (Fig. 6f, g). A similar increase in aggregates protein aggregation has previously been observed in a separate neuronal protein aggregation disease model in *Drosophila* following overexpression of Atg8a[60]. Beneficial or deleterious effects of increased Atg8a may reflect both the levels of the protein and the cellular context[61,62]. Specific mechanisms by which increased levels of Atg8a might promote cellular toxicity and dysproteostasis include defective cargo recognition[63], autophagy inhibition[64], or altered membrane fusion[65].

Our transcriptional analyses identified significant numbers of gene expression changes both in neurons and in glia (Supplementary Figs. 1, 2). Appl expression has traditionally been described as broadly neuronal[18]. Similarly, well-controlled studies using knockout mice have suggested that APP expression is substantially neuronal, with little to no detectable glial expression[66]. Thus, Appl may exert a previously unsuspected noncell autonomous effect not only on glial cells within the brain parenchyma, but also on adjacent organs such as the fat body (Supplementary Figs. 1, 2). The relatively small degree of overlap between genes with altered expression in non-neuronal cell types (Supplementary Fig. 2b, c, d) is also consistent with non-cell autonomous regulation by Appl, which would plausibly engage different mechanisms than those mediating cell-autonomous regulation. However, studies based on transgenic manipulation of Appl expression in glial cells have suggested a role for Appl in regulating glutamate recycling in glial cells[67], raising the possibility that Appl exerts a cell autonomous role in non-neuronal cells. Additional work defining the expression pattern of Appl at high resolution and sensitivity and manipulating Appl and downstream pathways separately in neurons and non-neuronal cells will be required to distinguish non-cell autonomous and cell autonomous functions of Appl.

Our studies demonstrate shortened lifespan and age-dependent neurodegeneration in flies lacking Appl function (Fig. 1, Supplementary Fig. 10). These findings are consistent with prior work on Appl by others[8] and are features of Alzheimer's disease in patients. In contrast, conditional triple knockout mice with selective inactivation of APP, APLP1 and APLP2 postnatally in excitatory neurons do not display clear neurodegeneration with age[55]. These disparate results may reflect differences in the timing or cellular specificity of gene knockout. Alternatively, dissimilarities in organismal biology such as reduced redundancy may unmask disease-relevant phenotypes in flies that are not apparent in mice. For example, flies lacking parkin or Pink1 function show significant cellular toxicity and mitochondrial dynamics defects[68–70] relevant to those seen Parkinson's disease patients, while even aged triple parkin/PINK1/DJ-1 mice do not show clear neurodegeneration[71]. Our demonstration of altered TFGβ signaling and proteostasis in double conditional knockout mice lacking APP and APLP2 function in neurons, and in human neurons lacking APP, suggest a conserved role for regulation of these pathways by

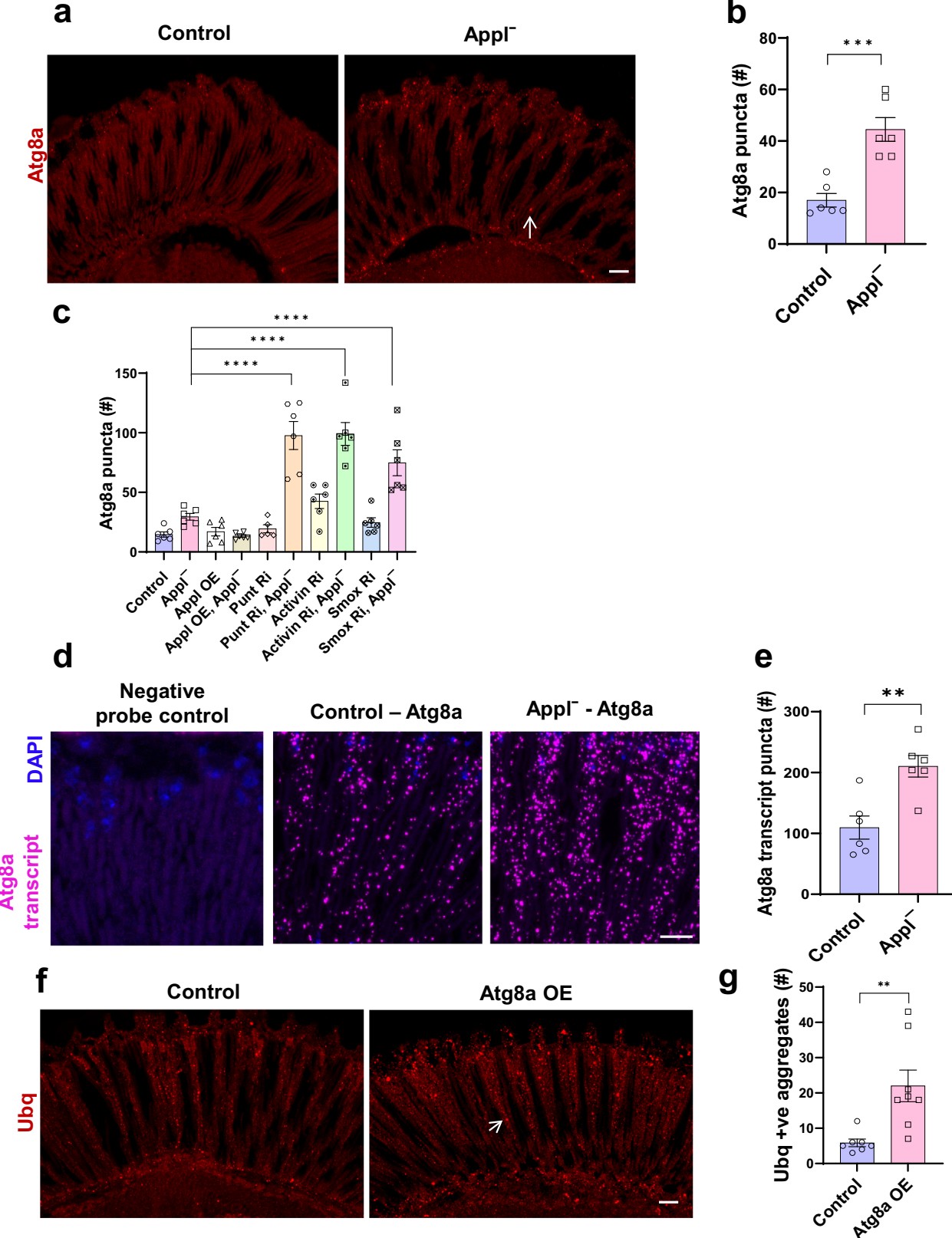

APP. Interestingly, evidence for dysregulation of neuronal TGFβ signaling has been seen in the brains of patients with Alzheimer's disease at postmortem examination[72–74]. Age-dependent loss of proteostasis itself has been strongly implicated as a contributing factor to the development of Alzheimer's disease and related neurodegenerative disorders[20].

We additionally explored the potential connection of our findings with Alzheimer's disease by expressing human transgenic tau in flies lacking Appl function. We observed enhancement of tau neurotoxicity with Appl deletion. Tau neurotoxicity is widely viewed as acting downstream of the toxicity of Aβ peptides in Alzheimer's disease[75]. Our findings raise the possibility that loss of Appl function might

**Fig. 6 | Appl loss alters the autophagy pathway. a** Representative immuno-fluorescent image shows increased numbers of Atg8a-positive puncta (arrow) in *Appl⁻* mutants compared to control flies. **b** Quantitative analysis shows significant increase in Atg8a-immunoreactive puncta in retinal sections from *Appl⁻* flies compared to controls. *p*-value (Control vs *Appl⁻*) = 0.0008. **c** Quantitative analysis indicates the number of Atg8a-immunoreactive puncta in retinal sections from flies expressing *Drosophila* Appl or transgenic *RNAi* directed to TGFβ components in retinal neurons. *p*-values for *Appl⁻* vs punt Ri, *Appl⁻*, *Appl⁻* vs Activin Ri, *Appl⁻*, and *Appl⁻* vs Smox Ri, *Appl⁻* are <0.0001. **d** Representative RNAi in situ (RNAScope) images show increased *Atg8a* mRNA (magenta) in the *Appl⁻* retina compared to control. DAPI (blue) shows nuclei. **e** Quantitative analysis shows significantly

increased *Atg8a* transcript levels in *Appl⁻* mutants compared to controls. *p*-value (Control vs *Appl⁻*) = 0.0031. **f** Representative immunofluorescent staining shows ubiquitin-positive aggregates (arrow) in flies overexpressing Atg8a (Atg8a OE) in neurons. **g** Quantitative analysis shows significantly increased numbers of ubiquitin-positive aggregates in sections of flies overexpressing Atg8a in retinal neurons. Control *n* = 7, *Appl⁻* *n* = 8. *p*-value (Control vs Atg8a OE) = 0.0079. Control is *nSyb-GAL4/+*. \*\**p* < 0.01, \*\*\**p* < 0.001, \*\*\*\**p* < 0.0001 two-tailed Student's t-test (**b**, **e**, **g**) or one-way ANOVA with Student-Newman-Keuls posthoc test (**c**). Data are represented as mean ± SEM. *n* = 6 per genotype (**b**, **c**, **e**). Scale bars are 10 µm. Flies are 10 days old. Source data are provided as a Source Data file.

alternatively or additionally promote cell death through mechanisms mediating toxicity of tau to neurons[49,50]. The concept of loss of APP function contributing to Alzheimer's disease has been supported by some studies demonstrating decreased APP transcript or protein levels in patient brains or CSF[76–78]. However, other studies have not reproduced these findings[79], raising the possibility that multiple mechanisms may contribute to disease pathogenesis in the complex context of sporadic Alzheimer's disease.

Given the preponderance of evidence, discussed above, that Alzheimer's disease associated mutations in APP influence cleavage of the protein it is tempting to speculate that cleavage might regulate TGFβ pathway activation by APP. We found that the extracellular domain of Appl was sufficient to rescue abnormal protein aggregation in flies lacking Appl function (Supplementary Fig. 6). It would be of interest to determine the effect of Alzheimer's disease promoting and protective mutations in APP on TGFβ activation and proteostasis in mammalian neurons.

Finally, our results may have implications for treatment strategies in Alzheimer's disease. Our work supports modulation of TGFβ signaling as a potential therapeutic target in the disorder, although the complexity of ligands, receptors, downstream signaling molecules and cell-type specific effects will require careful consideration. Depletion or deletion of APP has been suggested as a therapeutic option in Alzheimer's disease[80]. While the substantial redundancy of mammalian APP family members[3] indicates that therapeutic reduction in APP levels may be well tolerated, our findings suggest that the effects of removing or decreasing APP on TGFβ signaling and proteostasis should be monitored, and that the effects of reducing APP in the context of tauopathy should also be evaluated.

# Methods
## Ethics statement
This study complies with all relevant ethical regulations.

## *Drosophila* and mouse genetics
Flies were crossed and aged at 25˚C. They were reared in standard fly food containing yeast, agar, and corn flour. Pan-neuronal expression of transgenic RNAi and proteins was mediated by *elav-GAL4* in the genome scale screen and *nSyb-GAL4* in subsequent follow up studies. The genome scale screen was performed on flies aged for 30 days, as will be described in detail elsewhere (Leventhal, Fraenkel, Feany, in preparation). Transgenic flies expressing the 0N4R isoform of wild type or mutant human tau under the control of the UAS promotor have been described previously[15]. The following stocks were obtained from the Bloomington *Drosophila* Stock Center and Vienna *Drosophila* Resource Center: *nSyb-GAL4, elav-GAL4, UAS-punt-RNAi* (TRiP.GL00069, TRiP.HMS01944), *UAS-Actβ-RNAi* (TRiP.JF03276, GD3157), *UAS-Smox-RNAi* (TRiP.JF02320), *UAS-Atg8a-GFP, UAS-GFP-mCherry-Atg8a, Appl^tl, UAS-APP.695.Exel, UAS-Appl.s. UAS-CD8-PARP-Venus* was kindly provided by Darren Williams.

Behavior tests and lifespan assays were performed at 25˚C. Climbing was tested on day 10 or 20 using a standardized protocol as described here[16]. Briefly, flies were transferred to a vial without food

and gently tapped to begin the assay. The number of flies climbing 5 cm in 10 seconds was recorded. Approximately 10 flies were placed in each vial; 10 vials per genotype were assayed. The percentage of flies climbing was graphed for comparison. For lifespan assays flies were collected on day one following eclosion and the fly media was changed every three days. At least 350 flies were used per genotype.

Neuronal-APP conditional knockout mice were generated by crossing floxed APP mice with transgenic mice expressing Cre-recombinase under the neuronal nestin promoter. These neuronal conditional knockout mice were bred with APLP2 null mice to generate neuronal double conditional knockout (N-dCKO) mice as described in detail[38].

## Molecular biology
For genotyping of *Appl* deleted flies primers targeting the first exons (Forward - GCT GCG TCG TAA TTT GTT GC, Reverse - TCA CCT GAA CCT GAG CAG TG) and last exon (Forward - CGT CAC AAC ACA CCA TCC CA, Reverse - AGG TCG GAT TCT CGT ACC CA) were initially used. We confirmed that these exons were present in the *Appl^tl* flies. Next, we designed primers targeting the neighboring regions. Regions targeted by the following primers were intact in *Appl^tl* mutants: 5' end – Forward: CGG TTT TTG CAC TCG CTT GA, Reverse: AGC CGG ACA AAA GGA CAA CA, 3' end – Forward: ACA CTG AGT ATG GGG AGG CA, Reverse: CAA ATG CGG CAC GAG TTG AC. Regions targeted by the following primers were absent in *Appl* null flies: 5'end – Forward: GTC TGA TAT CGG GGG AAC CG, Reverse: CCA CAC AAA CGC ACT TCC AC. 3' end – Forward: CGG CAC CTA TTG AAC TCT GGA, Reverse: TCA TCG ACT GGT TTA CGG CT. Our results showed that the first two exons and the last three exons were present in the *Appl* deleted flies (Fig. 1c).

Primers for real-time PCR were selected from the *Drosophila* RNAi Screening Center (DRSC) FlyPrimerBank. The EcRB1 primers used were: Forward - GCA AGG GGT TCT TTC GAC G, Reverse - CGG CCA GGC ACT TTT TCA G. A total of 10 fly heads per condition were homogenized in Qiazol (Qiagen) and total RNA was isolated. Samples were treated with deoxyribonuclease and complementary DNA (cDNA) prepared using a High Capacity cDNA Reverse Transcription Kit (Applied Biosystems). Amplification was reported by SYBR Green in a QuantStudio 6 Flex (Thermo), and relative expression was determined using the ΔΔCt method normalized to the RPL32 housekeeping gene (Forward: GAC CAT CCG CCC AGC ATA C, Reverse: CGG CGA CGC ACT CTG TT).

## Histology, immunostaining and imaging
*Drosophila* were fixed in formalin at 10, 20, or 30 days of age, as specified in the results and figure legends, and embedded in paraffin or frozen for cryosectioning. Serial frontal sections (2 or 4 µm) of the entire brain were prepared from paraffin embedded material and mounted on glass slides. Sections were stained with hematoxylin and eosin to assess vacuole number. The number of vacuoles greater than 3 µm were counted throughout the entire fly brain. For immunostaining on paraffin sections, antigen retrieval was performed by microwaving slides in sodium citrate buffer for 15 minutes. For immunostaining on cryosections, tissue sections were fixed in 4%

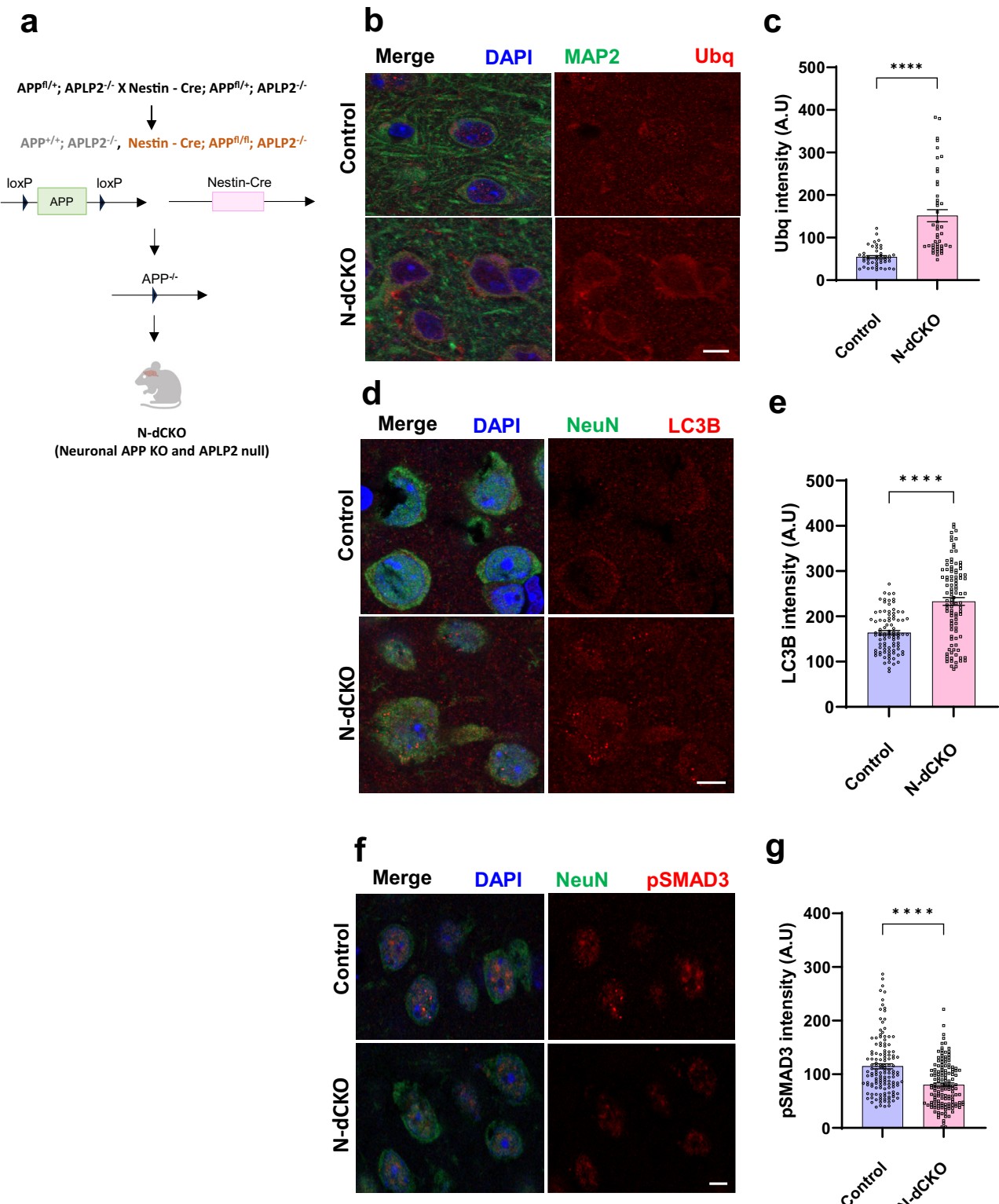

**Fig. 7 | Proteostasis defects in mice with loss of APP and APLP2 in neurons.**
**a** Schematic of crossing scheme used to obtain mice with *APLP2* knockout and neuronal conditional deletion of *APP*, termed N-dCKO. Created with BioRender.com. **b** Representative immunofluorescent images show increased ubiquitin (red) immunostaining in N-dCKO neurons (MAP2, green), additionally stained for DAPI (blue), compared to controls. **c** Quantitative analysis shows significantly increased ubiquitin levels in N-dCKO neurons compared to controls. Control $n = 45$, App KO $n = 45$. *p*-value = 2.61E-08. **d** Representative confocal images show increased staining for LC3B (red), a marker of autophagy, in N-dCKO neurons (NeuN, green) compared to controls. **e** Quantitative analysis shows a significant

increase in LC3B intensity in N-dCKO neurons compared to controls. Control $n = 90$, App KO $n = 100$. *p*-value = 4.98E-11. **f** Representative immunofluorescent images using an antibody specific for phosphorylated SMAD3, a TGFβ transcription factor, show decreased phospho-SMAD3 (red) in N-dCKO neurons (NeuN, green) compared to controls. **g** Quantification shows significantly decreased phospho-SMAD3 in N-dCKO neurons compared to controls. Control is *APLP2*$^{-/-}$. Control $n = 136$, App KO $n = 141$. *p*-value = 1.06E-08. ****$p < 0.0001$, two-tailed Student's t-test. Data are represented as mean ± SEM. $n = 5$ mice per genotype. Scale bars are 5 μm. Mice are 18 months old. Source data are provided as a Source Data file.

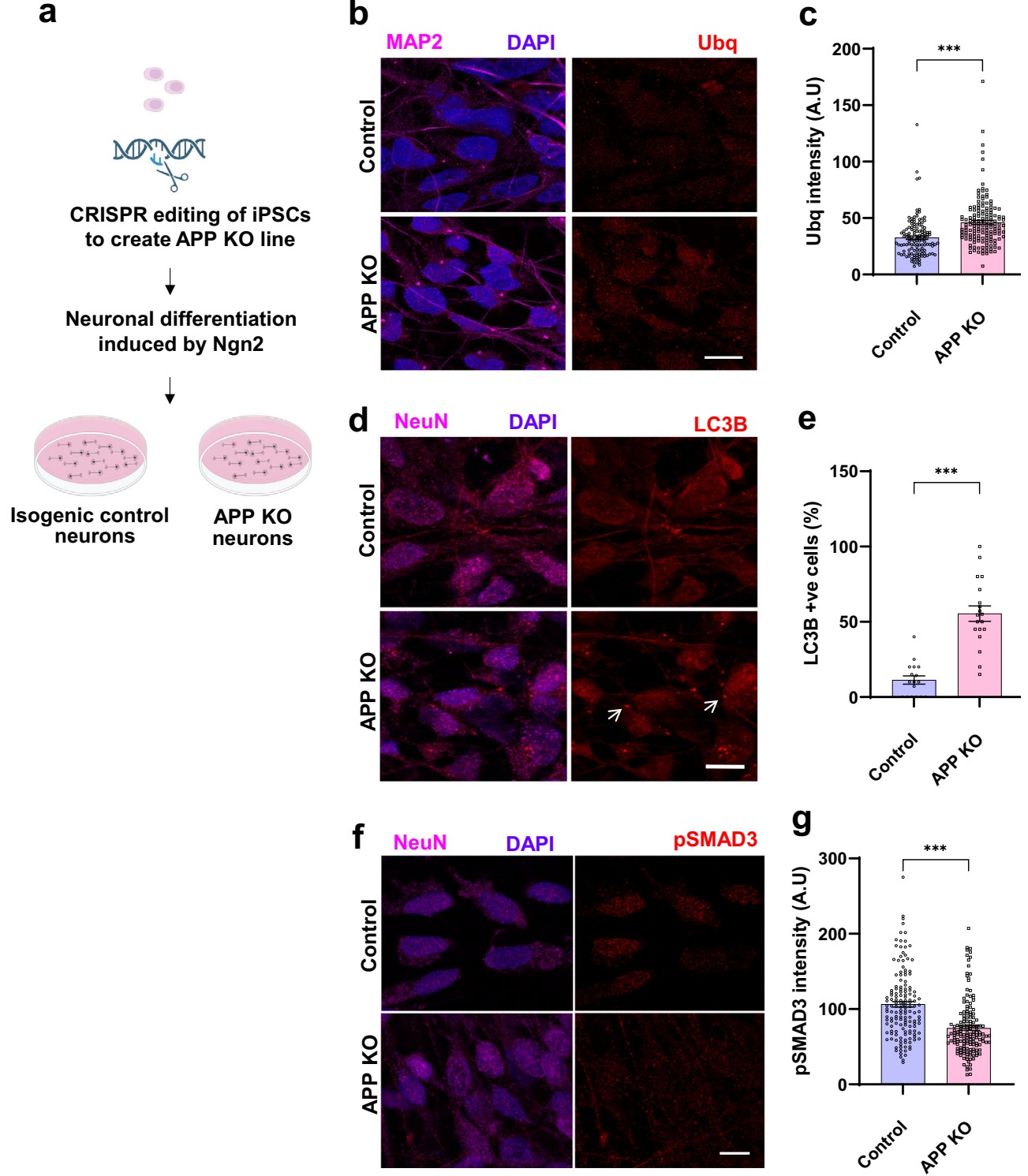

**Fig. 8 | Proteostasis defects in human iPSC-derived APP knockout neurons.**
**a** Schematic of the approach to generate APP knockout and isogenic control neurons. Created with BioRender.com. **b** Representative immunofluorescence images show increased ubiquitin (red) immunostaining in APP knockout neurons (MAP2, magenta), additionally stained for DAPI (blue), compared to controls. **c** Quantitative analysis shows significantly increased ubiquitin levels in APP knockout neurons compared to controls. Control $n = 133$, APP KO $n = 151$. $p$-value = 8.85E-09. **d** Immunostaining shows increased LC3B (arrow, red) immunostaining in APP knockout neurons (NeuN, magenta) compared to controls.

**e** Quantitative analysis shows significantly increased LC3B levels in APP knockout neurons compared to controls. Control $n = 17$, APP KO $n = 19$. $p$-value = 3.86E-08. **f** Immunostaining shows decreased phospho-SMAD3 (red) immunostaining in APP knockout neurons (NeuN, magenta) compared to controls. **g** Quantitative analysis shows significantly decreased phospho-SMAD3 levels in APP knockout neurons compared to controls. Control $n = 150$, APP KO $n = 156$. $p$-value = 9.76E-11. ***$p < 0.0001$, two-tailed Student's t-test. Data are represented as mean ± SEM. Cells from 3 independent differentiations of APP knockout and isogenic control cells were analyzed. Scale bars are 10 μm. Source data are provided as a Source Data file.

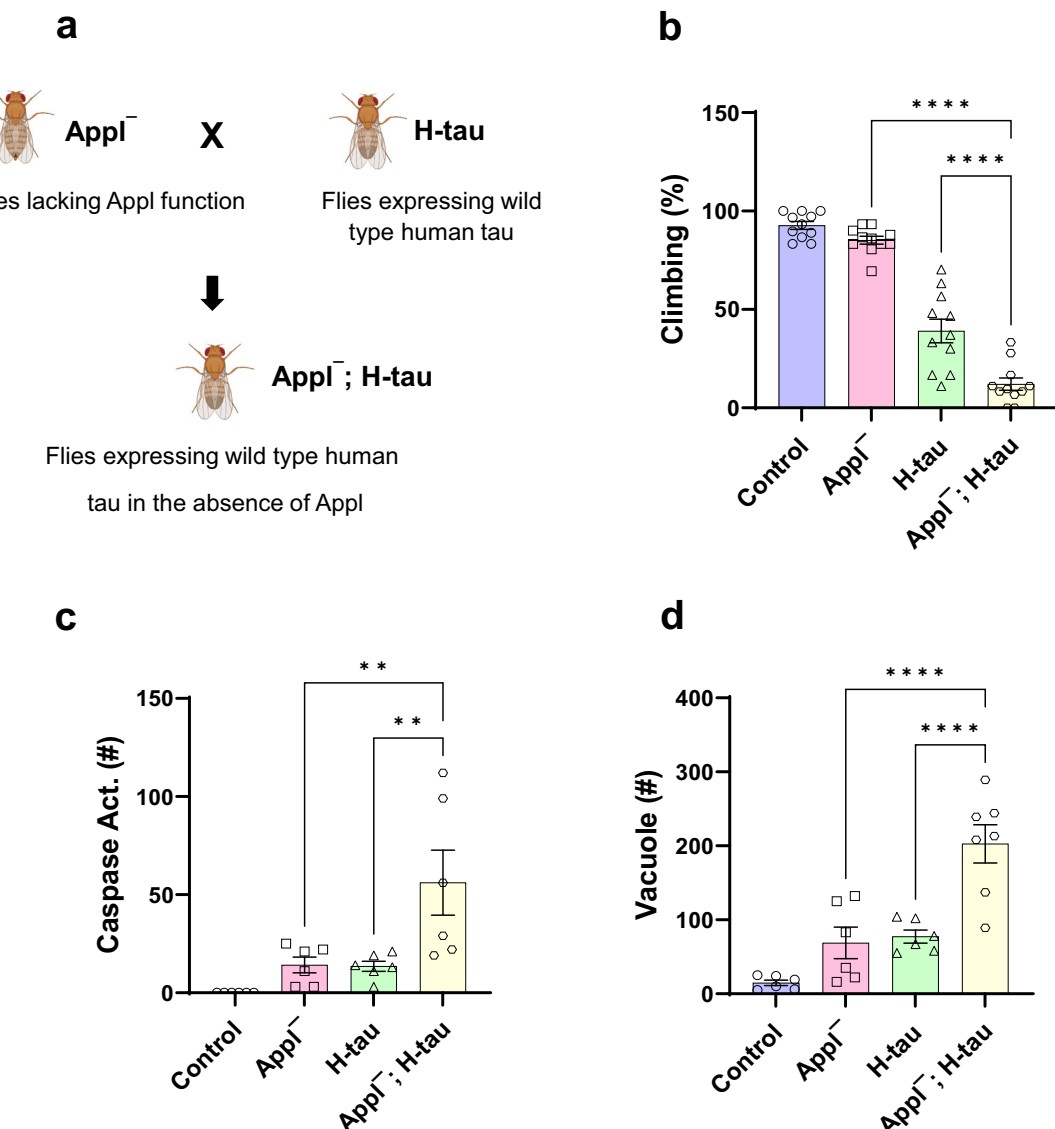

**Fig. 9 | Loss of Appl worsens tauopathy in *Drosophila*. a** Schematic of crossing scheme to obtain flies expressing wild type human tau in an *Appl⁻* mutant background. Created with BioRender.com. **b** Locomotor dysfunction in human tau transgenic flies is worsened by removing Appl function as monitored by the climbing assay. *p*-values for *Appl⁻* vs *Appl⁻* H-tau and H-tau vs *Appl⁻* H-tau are <0.0001. **c** Quantification shows increased numbers of neurons with caspase activation in whole brains of tau transgenic flies lacking Appl function. *p*-values for *Appl⁻* vs *Appl⁻* H-tau and H-tau vs *Appl⁻* H-tau are 0.002. **d** Quantitative analysis shows increased numbers of vacuoles in whole brains of tau transgenic flies lacking Appl function. *p*-values for *Appl⁻* vs *Appl⁻* H-tau and H-tau vs *Appl⁻* H-tau are <0.0001. Control is *nSyb-GAL4/+* in (**b**, **d**) and *UAS-CD8-PARP-Venus, nSyb-GAL4/+* in (**c**). **p < 0.01, ****p < 0.0001, one-way ANOVA with Student-Newman-Keuls posthoc test. Data are represented as mean ± SEM. In **b**, n = 11 repeats per genotype, Control = 84, *Appl⁻* = 109, H-tau = 106, *Appl⁻* H-tau = 75 flies. In **c**, n = 6 flies per genotype. In **d**, Control = 6, *Appl⁻* = 6, H-tau = 6, *Appl⁻* H-tau = 7 flies. Flies are 20 days old. Source data are provided as a Source Data file.

paraformaldehyde for 15 minutes. Slides were blocked in PBS with 2% milk or BSA in PBS and 0.3% Triton X-100, followed by overnight incubation with primary antibodies at room temperature. The manufacturer's protocol was followed for staining with the Enzo Life Sciences ProteoStat dye. Briefly, paraffin sections were baked at 60 °C for 10 minutes. After processing through xylene and alcohol, tissue sections were fixed using 4% paraformaldehyde. Slides were then incubated with ProteoStat dye followed by destaining in acetic acid. At least 6 brains were examined per genotype and time point for *Drosophila* histology and immunostaining. Flies expressing GFP-mCherry-Atg8a were dissected in Schneider's medium and imaged immediately using confocal microscopy as described[34,81].

Brains from 18-month-old mice were postfixed in 4% paraformaldehyde, infiltrated with 30% sucrose and sectioned at 40 μm. Free-floating vibratome sections were permeabilized using PBS with

0.1% Triton X-100, blocked with PBS with 0.1% Triton X-100 and 3% BSA for 1 hour at room temperature, and incubated in primary antibody overnight at 4 °C. Five brains per genotype were examined.

Cultured neurons were fixed in 100% methanol, washed, blocked in PBS with 0.1% Triton X-100 and 5% BSA, and incubated in primary antibody overnight at 4 °C. At least three independent differentiations of APP knockout and isogenic control neurons plated in parallel were performed and analyzed. Three coverslips were analyzed for each differentiation.

Primary antibodies to the following proteins were used at the indicated concentrations: ubiquitin (1:1000, P4G7, BioLegend; 1:200 Cell Signaling; 1:200, Z0458, Dako), GABARAP (1:1000, endogenous *Drosophila* Atg8a, E1J4E, Cell Signaling; 1:200, Abcam), LC3B (1:200, Novus Biologicals), ref(2)P (1:600, Sarkar et al.[34],), Rab5 (1:100, ab31261, Abcam), Rab11 (1:100, 610656, BD Biosciences), GFP (1:200,

N86/8, NeuroMab), cleaved PARP (1:5000, E51, Abcam), elav (1:5, 9F8A9, Developmental Studies Hybridoma Bank), NeuN (1:400, EMD Millipore), MAP2 (1:100, EMD Millipore), pSMAD3 (1:200, Abcam). Appropriate secondary antibodies coupled to Alexa Fluor 488 or Alexa Fluor 555 were incubated for one to two hours at room temperature.

Fluorescence imaging was performed with laser-scanning confocal microscopy using a Zeiss LSM-800 and a 63X objective lens. Fluorescence from varying depths was captured as a z-stack. Images were processed in ZEN Blue software. The same settings were used for scanning of both control and test sections. Quantification of ubiquitinated protein aggregates in *Drosophila* retinas was performed by counting the number of ubiquitin-immunoreactive puncta in well oriented retinal sections. RNA in situ hybridization results were quantified in retinal sections using the analyze particles plugin in ImageJ. At least 6 animals were examined per genotype and time point for analyses of *Drosophila* retinas. To assay autophagic flux, *Drosophila* brains expressing GFP-mCherry-Atg8a were examined. Quantification of ratio between GFP and mCherry in *Drosophila* brain was performed using ImageJ as described[34,81]. For immunofluorescence of mouse and human neurons the mean fluorescence intensity was quantified using ImageJ. At least 45 neurons were analyzed per genotype in mouse sections. Five mouse brains were examined per genotype. Quantitative imaging of cultured neurons was performed on at least 100 neurons per genotype, representing 3 independent differentiations per genotype and 3 coverslips per differentiation.

## Single-cell sequencing
Flies were collected and aged for 10 days and brains were then dissected from heads. Three replicates were used for each genotype. Twenty male and female flies were used for each replicate. A single-cell suspension was made from the brains by following mechanical and enzymatic digestion using trypsin-EDTA[82,83]. Using acridine orange/ propidium iodide stain, cells were quantified in the Luna fluorescence cell counter. Single-cell libraries were generated using 10X genomics chromium next GEM single cell 3′ kit. Single-cell sequencing was performed using NovaSeq 6000. Quality control was performed by removing cells with a low number of genes and a higher (>10%) percent of mitochondrial counts. Data were visualized using uniform manifold approximation and projection (UMAP). Seurat, R package, was used for cell clustering. Based on the marker genes in the DRSC scRNA-seq database, cell clusters were annotated[82,84]. Gene ontology was performed using gene ontology (www.geneontology.org)[85] and FlyEnrichr[86]. The GO library was generated based on the information from the gene ontology consortium. Genes were categorized under biological processes and molecular functions. GO terms were ranked based on the combined score calculated by multiplying the $p$-value and z-score using the formula $c = \ln(p) * z$. The $p$-value is calculated by Fisher's exact test or the hypergeometric test. The z-score computes the deviation from the expected rank using Fisher's exact test.

## Proteomics and ubiquitinomics
Flies were collected and aged for 10 days. An equal number of males and females were used in the study. Six replicates for control and five replicates for *Appl* null were performed. For each replicate, 550 flies were used. Initially, heads were isolated and lysed in a lysis buffer containing 8 M urea. Protein concentration was measured using bicinchoninic acid (BCA) assay. Samples were reduced, and alkylated, followed by precipitation, resuspension, and trypsin digestion. 35 μg protein per channel was used for whole proteomics, and 2 mg per channel was used for ubiquitinomics. Samples were TMT labeled, fractionated, and liquid chromatography–mass spectrometry was performed. Ubiquitinomics data was normalized with total TMT signal and nonredundant ubiquitinated sites were taken for further analysis.

## Network analysis
Pathway enrichment was assessed by the hypergeometric test used in gProfiler. Proteomics with an FDR < 0.0001 and ubiquitinomics with an FDR < 0.1 were integrated with OmicsIntegrator2. Clusters were separated using Louvain clustering. The reference interactome was the *Drosophila* STRING protein-protein interactome limited to experimentally validated edges. Node weights were calculated as the negative $\log_{10}$ FDR-adjusted $p$-value between mutant and control for the proteomics and ubiquitinomics. Results from single-cell RNA sequencing data were used to adjust edge confidence and node weights in the network to prioritize genes differentially expressed across many cell types. Weights for the input proteomics and ubiquitinomics were adjusted by multiplying the negative $\log_{10}$ FDR-adjusted $p$-value by the proportion of number of differentially expressed single-cell RNA-seq clusters to the average number of single cell RNA-seq clusters. To adjust the edge costs, we calculated the average ratio of number of differentially enriched clusters to the average number of clusters for the two nodes involved in the edge. We then calculated the rank order of these computed weights and divided the rank order by the number of edges. We multiplied this new weight by the edge cost. Lower cost in the network shows higher confidence.

## RNAscope
The RNAscope assay was performed by following the protocol provided by the manufacturer supplied in the ACD HybEZ II hybridization systems user manual. Briefly, formalin-fixed and paraffin-embedded heads were sectioned and processed through xylene and ethanol. Sections were permeabilized and hybridized with a gene-specific probe. A probe targeting residues 359-1186 of the *Atg8a* mRNA was used. Hybridization-based signal amplification and noise reduction were performed. Opal fluorophore 650 was used for visualization under confocal microscopy.

## Induced pluripotent stem cells (iPSCs) and neuronal differentiation
Fibroblasts from fetal lungs were reprogrammed to iPSCs. CRISPR editing with the gRNA sequence GCTGCAGCGAGACCTACCCG was used to create the APP knockout cell line. Isogenic APP knockout and control NGN2 inducible iPSCs were obtained from Brigham and Women's iPSC Neurohub. Cultures were maintained as feeder-free cells in a defined, serum-free media (mTeSR, STEMCELL Technologies). Neuronal induction was performed using a modification of a previously described protocol as elaborated here[44]. Cells were dissociated with Accutase (STEMCELL Technologies) and plated in mTeSR supplemented with 10 mM ROCK inhibitor Y-27632 and 2 mg/ml doxycycline on a Matrigel-coated 6-well plate. On day 1 of the differentiation, culture media was changed to DMEM/F12 supplemented with N2 (Invitrogen), B27 (Invitrogen), nonessential amino acids, GlutaMAX, 5 mg/ ml puromycin, and 2 mg/ml doxycycline. On day 4 of differentiation, media was changed to Neurobasal media (Invitrogen) supplemented with B27 (Invitrogen), 10 ng/ml BDNF, CNTF and GDNF, 10 mM ROCKi, 5 mg/ml puromycin, and 2 mg/ml doxycycline. Medium was changed every 3 days.

## Western blotting
Neurons were collected in PBS and lysed in Laemmli buffer. Fly heads were homogenized in Laemmli buffer. Biochemical isolation of insoluble protein aggregates was performed using Triton lysis buffer (50 mM Tris-HCl pH 7.4, 1% Triton X-100, 150 mM NaCl, 1 mM EDTA) and the pellet was resuspended in 4% SDS containing lysis buffer[21,24,25]. Samples were run on 4-20% gels and transferred to nylon membranes using the Bio-Rad Trans-Blot Turbo Transfer System. Antigen retrieval was performed on membranes by boiling in PBS followed by blocking in PBS with 3% milk and 0.05% Tween 20. At least three independent differentiations of triplication and isogenic control neurons plated in

parallel were performed and analyzed. Primary antibodies were incubated at 4 °C overnight. Primary antibodies to the following proteins were used at the indicated concentrations: Ubq (1:5000, P4D1, Cell Signaling; 1:5000, P4G7-HRP, BioLegend), Actin (1:10,000, Developmental Studies Hybridoma Bank), pSMAD3 (1:5000, Abcam), SMAD3 (1:5000, Abcam), pSMAD1/5/9 (1:2000, Cell Signaling), GABARAP (1:2000, Abcam), APP (1:1000, Sigma). PHF1 (1:50,000, gift from Peter Davies), AT8 (1:10,000, Thermo), AT180 (1:50,000, Thermo), AT270 (1:10,000, Thermo), total tau (1:75,000, A0024, Dako), GAPDH (1:20,000, Invitrogen). HRP-conjugated secondary antibodies were applied for 3 hours and signal was detected using enhanced chemiluminescence.

## Statistical analysis

Two-tailed t-tests were used to compare two groups. One-way ANOVA or two-way ANOVA followed by the Student-Newman-Keuls test were used to compare multiple samples. Statistical analyses with a $p$-value less than 0.05 were considered significant. Bar graphs are represented as mean ± standard error of measurement (SEM). Excel and GraphPad prism were used for statistical analyses. Statistical tests, $p$-values and number of replicates are specified in the legend of each figure.

## Reporting summary

Further information on research design is available in the Nature Portfolio Reporting Summary linked to this article.

## Data availability

The proteomics and ubiquitinomics data generated in this study have been deposited in the ProteomeXchange Consortium under accession code PXD041862. The single-cell RNA sequencing data generated in this study have been deposited in the Gene Expression Omnibus (GEO) repository under accession code GSE231518 [http://www.ncbi.nlm.nih.gov/geo/]. Publicly available DRSC scRNA-seq DataBase [https://www.flyrnai.org/tools/single_cell/web/] was used for annotation of the cell clusters. Source data are provided with this paper.

## Code availability

All code used in this study is publicly available at https://github.com/bwh-bioinformatics-hub/HUB_vanitha2022_scRNA-seq.

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

## Acknowledgements

Fly stocks obtained from the Vienna *Drosophila* Resource Center, Bloomington *Drosophila* Stock Center (NIH P40OD018537) and D. Williams were used in this study. We thank the Transgenic RNAi Project (TRiP) at the Harvard Medical School (NIH-NIGMS R01GM084947) for making transgenic RNAi stocks. Single cell RNA sequencing was performed by the Single Cell Core and mass spectrometry was performed by the Thermo Fisher Scientific Center for Multiplexed Proteomics at Harvard Medical School, Boston, MA. Monoclonal antibodies were obtained from the Developmental Studies Hybridoma Bank developed under the auspices of the NICHD and maintained by the University of Iowa, Department of Biology, Iowa City, IA 52242. Dr. Hui Zheng kindly provided the APP/APLP2 neuronal double conditional knockout mouse brain sections. Dr. Tracy Young-Pearse contributed the APP knockout and control cell lines. Dr. Peter Davies generously provided the PHF1 antibody. We thank Yi Zhong for excellent technical assistance. Diagrams were created with BioRender.com. This research was funded in whole or in part by Aligning Science Across Parkinson's [ASAP-000301] through the Michael J. Fox Foundation for Parkinson's Research (MJFF), NIH-NIA R01AG33518, R01AG057331, R01AG057294, and by the Ellison Medical Foundation. For the purpose of open access, the author has applied a CC BY public copyright license to all Author Accepted Manuscripts arising from this submission.

## Author contributions

V.N. and M.B.F. designed the study, performed experiments and wrote the manuscript. H.B., M.J.L., R.A.B., X.D. and E.F. performed experiments or analyzed data and edited the manuscript. M.B.F. and E.F. obtained funding for the research.

## Competing interests

The authors declare no competing interests.
