## [Peer Review File · Nature Communications]

Integrative analysis reveals a conserved role for the amyloid precursor protein in proteostasis during agingREVIEWER COMMENTS

Reviewer #1 (Remarks to the Author):

In the current manuscript, from a *Drosophila*-based forward genetic screen, Nithianandam et al., identified Appl as a protein required to maintain neuronal viability. By performing transcriptomic and proteomic analysis of Appl mutant flies and further mechanistic studies not only in flies but also in mouse model and human neurons the authors established that Appl has a key role in control of proteostasis via the TGFbeta signaling.

Overall, this is a well-developed study with an interesting and robust conclusion obtained from the analysis of multiple disease models. The idea that APP loss of function could be a contributor of AD pathogenesis via proteostasis impairment is novel and significant. The experiments are well described, and the results support the conclusions and claims. In my opinion this study is suitable for publication. Nevertheless, I have two minor comments that the authors could address in a revised version.

1. When APP was knockout in human iPSC-derived neurons, the authors confirmed the efficiency of the knockout by APP Western blot (Supplementary Figure 6). Did the authors check for APLP1 and APLP2 in these cells to see if there is any compensatory mechanism in iPSC as observed in APP KO mice?

2. The authors suggested that loss of proteostasis function of APP could be relevant for Alzheimer's Disease. It would be important to include in the discussion a paragraph that could support the idea of a possible Loss of function of APP in AD. For example, how are the levels of APP mRNA in AD human brains versus controls? There are several APP cleavage products that are not related to amyloid formation (C83, C99, AICD or sAPPalpha). There is any difference in the levels of this cleavage products in AD versus control? Could any of these cleavage products have a role in proteotasis? How are the levels of APLP1 and APLP2 in AD brain samples versus controls?

Reviewer #2 (Remarks to the Author):

This manuscript describes a new function of the APP protein and its fly homologue APPL in regulating autophagy. This is an important result due to changes in APP being related to Alzheimer's disease but while there has been a focus on A β , other functions of the protein have not been well studied. That APP, and as shown in the manuscript specifically is extracellular part, plays a role in autophagy is therefore important for a better understanding of the pathology. The authors show that this function of APP

proteins is conserved from flies to humans and therefore relevant for the disease. The work is original, the experiments are well performed, and the results support the conclusions. I do not have concerns that need to be addressed before publication.

Reviewer #3 (Remarks to the Author):

Summary

In this manuscript, Nithianandam et al. sought to examine the function of APP in health and disease states. Using *Drosophila* as a model system the authors found that neuronal depletion of the only APP family member (*Appl*) evokes a neurodegenerative phenotype. This finding was independently confirmed by detection of increased caspase-3 activation and formation of vacuoles in brain sections from mutant flies lacking *Appl*. Subsequently, the authors performed single-cell RNA sequencing of wild-type and *Appl* mutant fly brain and found an enrichment of genes annotated with mitochondria and protein synthesis among the up- and downregulated genes. Moreover, the authors identified TGFbeta/BMP signaling as pathway specifically affected in neuronal cells. Next, the authors performed protein abundance and ubiquitination site profiling and found proteostasis-relevant proteins altered. Consistently, the authors detected an increase in ubiquitin-positive structures in *Appl* mutant retinas. Focusing on the link between proteostasis regulation and TGFbeta/BMP signaling the authors detected altered TGFbeta/BMP signaling upon loss of *Appl*. Since TGFbeta signaling has already been linked to autophagy, the authors survey this pathway and found increased Atg8 and ref(2)P puncta. Lastly, the authors performed experiments to confirm their findings in mouse and iPSC models as well as a fly tauopathy model. Together, the work of Feany and colleagues provides evidence that loss of *Appl* affects cellular proteostasis. While this finding is potential interesting, the study is mostly descriptive with very little mechanistic insights.

At this preliminary stage the authors leave many questions unanswered. For example: How does *Appl* modulates TGFbeta signaling at the molecular level? Is *Appl* physically interacting with components of this pathway? Which part of *Appl* is mediating the interaction with this pathway? Where does this regulation take place in the cell? Are other signaling pathways affected in a similar way? How did TGFbeta signaling and autophagy components score in the proteomics analysis? Are any of the protein folding factors that were found upregulated by proteomics present in ubiquitin and/or Atg8a positive puncta?

Other major points are:

- 1) Is ongoing processing of *Appl* required for its proteostasis controlling function? The authors should perform rescue experiment with a non-processable variant of *Appl*.

2) The fact that downregulation of TGFbeta/BMP components does not significantly change the appearance of ubiquitin-positive structures in the presence of functional Appl argues against a prominent role of this signaling pathways in controlling proteostasis. Moreover, if TGFbeta/BMP signaling and Appl are actually acting in the same pathway, one would not expect an additive effect upon ablation of both components (as e.g. observed for Appl vs Appl/punt or any of the other combinations).

3) The increase in ubiquitin-positive structures is not necessarily a sign of altered proteostasis. This could also arise from mis-trafficking within the endocytic system (e.g. from plasma membrane to early endosomes or back via recycling endosomes). The authors need to monitor additional proteostasis markers which are not linked to endosomal defects to support their claim of a proteostasis imbalance. Along the same lines, the authors need to show that loss of Appl actually increases protein aggregation by analyzing SDS-insoluble fractions. Also, it would be very informative to determine which proteins are found in these aggregates.

4) Are the ubiquitin-positive structures also positive for Atg8 and p62? The authors need to perform triple or pair-wise costainings of ubiquitin, ATG8 and p62 in Appl WT and KO flies.

5) The increased puncta formation of Atg8 and ref(2)P is not necessarily a sign of altered autophagy. The authors need to perform autophagy flux assays to make such claims.

6) Should overexpression of Atg8a not lead to clearance of ubiquitin-positive structures? It is not clear what process the increase in ubiquitin puncta upon Atg8a overexpression reflects. Is Atg8a co-aggregating, is it decorating perturbed endosomal membranes or is Atg8a helping to form autophagosomes? But if the latter is the case, why are ubiquitin puncta not reduced?

Reviewer #1 (Remarks to the Author):

In the current manuscript, from a Drosophila-based forward genetic screen, Nithianandam et al., identified Appl as a protein required to maintain neuronal viability. By performing transcriptomic and proteomic analysis of Appl mutant flies and further mechanistic studies not only in flies but also in mouse model and human neurons the authors established that Appl has a key role in control of proteostasis via the TGFbeta signaling.

Overall, this is a well-developed study with an interesting and robust conclusion obtained from the analysis of multiple disease models. The idea that APP loss of function could be a contributor of AD pathogenesis via proteostasis impairment is novel and significant. The experiments are well described, and the results support the conclusions and claims. In my opinion this study is suitable for publication. Nevertheless, I have two minor comments that the authors could address in a revised version.

1. When APP was knockout in human iPSC-derived neurons, the authors confirmed the efficiency of the knockout by APP Western blot (Supplementary Figure 6). Did the authors check for APLP1 and APLP2 in these cells to see if there is any compensatory mechanism in iPSC as observed in APP KO mice?

APP knockout iPSC neurons do not have altered APLP1 or APLP2 levels.¹ APP knockout neuronal precursor cells do have elevated APLP1 and APLP2 mRNA and protein. However, following differentiation into neurons, there are no detectable differences in APLP1 or APLP2 levels.¹ We have now incorporated these findings into our manuscript.

2. The authors suggested that loss of proteostasis function of APP could be relevant for Alzheimer's Disease. It would be important to include in the discussion a paragraph that could support the idea of a possible Loss of function of APP in AD. For example, how are the levels of APP mRNA in AD human brains versus controls? There are several APP cleavage products that are not related to amyloid formation (C83, C99, AICD or sAPPalpha). There is any difference in the levels of this cleavage products in AD versus control? Could any of these cleavage products have a role in proteostasis? How are the levels of APLP1 and APLP2 in AD brain samples versus controls?

Reduced levels of APP have been reported in some studies of brains from patients with Alzheimer's disease compared to controls.^{2,3} Interestingly, when multiple brain regions were analyzed in these studies, levels of APP were more strongly reduced in preferentially vulnerable brain regions in Alzheimer's disease, including hippocampus and temporal cortex. Other studies have shown that levels of APP cleavage products are altered in Alzheimer's disease patients. In some studies secreted APP-alpha (sAPPalpha), considered neuroprotective, was reduced in the CSF of Alzheimer's disease patients.⁴ These findings are consistent with our observation that secreted Appl can rescue proteostasis defects in flies lacking Appl. Additional cleavage products of APP, such as AICD⁵ and C99,⁶ have also been reported to increase in patient brains. Some studies have suggested that levels of APLP2 are decreased in Alzheimer's disease patient brains³ while APLP1 has been shown to increase in CSF of patients with mild cognitive impairment.⁷ However, it is important to note that different studies of full length APP proteins and cleavage products in Alzheimer's disease have not always produced consistent results,⁸ perhaps reflecting the heterogeneity of the disease and challenges of studying postmortem material. We have therefore been circumspect in our discussion of this important point.

Reviewer #2 (Remarks to the Author):

This manuscript describes a new function of the APP protein and its fly homologue APPL in regulating autophagy. This is an important result due to changes in APP being related to Alzheimer's disease but while there has been a focus on A β , other functions of the protein have not been well studied. That APP, and as shown in the manuscript specifically is extracellular part, plays a role in autophagy is therefore important for a better understanding of the pathology. The authors show that this function of APP proteins is conserved from flies to humans and therefore relevant for the disease. The work is original, the experiments are well performed, and the results support the conclusions. I do not have concerns that need to be addressed before publication.

We thank the reviewer for their positive evaluation of our study.

Reviewer #3 (Remarks to the Author):

Summary

*In this manuscript, Nithianandam et al. sought to examine the function of APP in health and disease states. Using *Drosophila* as a model system the authors found that neuronal depletion of the only APP family member (*Appl*) evokes a neurodegenerative phenotype. This finding was independently confirmed by detection of increased caspase-3 activation and formation of vacuoles in brain sections from mutant flies lacking *Appl*. Subsequently, the authors performed single-cell RNA sequencing of wild-type and *Appl* mutant fly brain and found an enrichment of genes annotated with mitochondria and protein synthesis among the up- and downregulated genes. Moreover, the authors identified TGF β /BMP signaling as pathway specifically affected in neuronal cells. Next, the authors performed protein abundance and ubiquitination site profiling and found proteostasis-relevant proteins altered. Consistently, the authors detected an increase in ubiquitin-positive structures in *Appl* mutant retinas. Focusing on the link between proteostasis regulation and TGF β /BMP signaling the authors detected altered TGF β /BMP signaling upon loss of *Appl*. Since TGF β signaling has already been linked to autophagy, the authors survey this pathway and found increased Atg8 and ref(2)P puncta. Lastly, the authors performed experiments to confirm their findings in mouse and iPSC models as well as a fly tauopathy model. Together, the work of Feany and colleagues provides evidence that loss of *Appl* affects cellular proteostasis. While this finding is potential interesting, the study is mostly descriptive with very little mechanistic insights.*

*At this preliminary stage the authors leave many questions unanswered. For example: How does *Appl* modulates TGF β signaling at the molecular level? Is *Appl* physically interacting with components of this pathway? Which part of *Appl* is mediating the interaction with this pathway? Where does this regulation take place in the cell? Are other signaling pathways affected in a similar way? How did TGF β signaling and autophagy components score in the proteomics analysis? Are any of the protein folding factors that were found upregulated by proteomics present in ubiquitin and/or Atg8a positive puncta?*

The reviewer raises a number of interesting and important points. We have shown that expressing the extracellular region of *Appl* can rescue phenotypes associated with loss of *Appl* function (Supplementary Fig. 6a-c). Based on these results and prior literature demonstrating a direct physical interaction between the extracellular domain of APP and TGF β ligand^{9,10} we hypothesize that *Appl* may bind to the extracellular domain of TGF β ligand, blocking receptor activation and thus reducing downstream phosphorylation of the Smox transcription factor (Fig. 5). We have revised our manuscript to describe our findings and model more clearly. As we

demonstrate in the Table, TGF β signaling was the top candidate pathway from our transcriptional profiling. Nonetheless, we wondered, as the reviewer asks, how the proteomic and cell biological data addressed the role TGF β signaling, and whether other signaling pathways could be involved. To explore the latter question we obtained candidate RNAi and overexpression reagents for other signaling pathways identified either in the transcriptional data or as candidates from the literature. We then manipulated these pathways in flies lacking Appl function and in controls. TGF β signaling emerged as the top candidate from these genetic studies, without clear and consistent evidence for involvement of other signaling pathways. However, because our studies of other signaling pathways were preliminary in nature we cannot rigorously exclude a potential role for these pathways. Due to low levels of expression, as is common with signaling pathway components, we did not detect most TGF β signaling proteins in our proteomic studies. We did detect Smox and saw no difference between Smox levels in Appl mutants and controls, consistent with the well-documented regulation of Smox by phosphorylation, not total protein levels (Figs. 5,8, Supplementary Fig. 9). Importantly, Atg8a levels were increased in our proteomics data (Supplementary Fig. 6d), consistent with increased transcript levels of Atg8a in our RNAscope analysis (Fig. 6d, e). We also observed elevated levels of Atg4a in our proteomic analysis (Supplementary Table). Using available antibodies we were not able to detect Hsp27 in ubiquitin-positive aggregates. Antibodies were not available to other chaperones identified in our proteomics study.

Other major points are:

1) Is ongoing processing of Appl required for its proteostasis controlling function? The authors should perform rescue experiment with a non-processable variant of Appl.

This is a good experiment and one that we attempted to perform. Unfortunately, expression of the secretion-deficient variant of Appl (Appl-sd) was toxic under our experimental conditions and we did not recover viable animals for study. Neuronal toxicity of the non-processable variant of Appl in neurons has been observed previously.¹¹

2) The fact that downregulation of TGFbeta/BMP components does not significantly change the appearance of ubiquitin-positive structures in the presence of functional Appl argues against a prominent role of this signaling pathways in controlling proteostasis. Moreover, if TGFbeta/BMP signaling and Appl are actually acting in the same pathway, one would not expect an additive effect upon ablation of both components (as e.g. observed for Appl vs Appl/punt or any of the other combinations).

The reviewer makes an interesting point. We had to use partial loss of function TGF β reagents to assess genetic interactions between Appl and the TGF β system because complete loss of TGF β signaling is lethal. Since our genetic modifier lines only partially reduced levels of TGF β signaling components we were not able to determine the effect of complete loss of TGF β function on proteostasis in the aging adult brain. Nor were we able to evaluate epistasis in a classical fashion, which relies on analysis of null alleles.^{12,13} Our in vivo genetic data do show that reducing TGF β signaling has a synergistic effect with Appl loss in modulating proteostasis, arguing for a biological interaction between the two.¹³ Our biochemical data further demonstrate that activity of the pathway and downstream targets are altered in flies lacking Appl function. We thus provide multiple lines of evidence linking Appl and TGF β signaling with proteostasis dysfunction during aging and suggest that altering TGF β signaling may be an effective method of promoting normal proteostasis in the context of impaired proteostasis due to loss of Appl function.

3) The increase in ubiquitin-positive structures is not necessarily a sign of altered proteostasis.

This could also arise from mis-trafficking within the endocytic system (e.g. from plasma membrane to early endosomes or back via recycling endosomes). The authors need to monitor additional proteostasis markers which are not linked to endosomal defects to support their claim of a proteostasis imbalance. Along the same lines, the authors need to show that loss of Appl actually increases protein aggregation by analyzing SDS-insoluble fractions. Also, it would be very informative to determine which proteins are found in these aggregates.

Thank you for these helpful suggestions. We have now used additional methods to document the presence of insoluble protein aggregates in flies lacking Appl function, including demonstrating increased biochemically insoluble protein (Supplementary Fig. 3c,d) and staining with the commonly used ProteoStat dye (Supplementary Fig. 3a,b), which binds specifically to aggregated protein.¹⁴ In addition, we performed immunostaining with antibodies directed to the early endosomal small GTPase Rab5 and the recycling endosome marker Rab11. Neither endosome marker colocalized with ubiquitin-positive aggregates in flies lacking Appl function (Supplementary Fig. 3e,f). We agree that it would be of interest to determine the protein composition of age-related protein aggregates. Although the formation of ubiquitinated aggregates is pervasive with advancing age and in many age-related disease states, the protein composition of these aggregates is mostly uncharacterized. Aside from a few well-known examples, including beta-amyloid in amyloid plaques and tau in neurofibrillary tangles, the proteins present in aging-related ubiquitinated aggregates have been difficult to catalog, even in mammalian systems where much more starting material is available for biochemical analysis, at least partly reflecting the difficulty of purifying these aggregates.¹⁵

4) Are the ubiquitin-positive structures also positive for Atg8 and p62? The authors need to perform triple or pair-wise costainings of ubiquitin, ATG8 and p62 in Appl WT and KO flies.

We have performed the co-immunostaining experiments suggested. Many ubiquitin-positive inclusions that do not colocalize with Atg8a, suggesting that ubiquitin-positive aggregates are not simply abnormal autophagosomes. We do observe some puncta that stain for both Atg8a and ubiquitin (Supplementary Fig. 4c). Occasional ubiquitin-positive aggregates similarly colocalize with p62 (Supplementary Fig. 4d). Colocalization of Atg8a and ref(2)P with ubiquitinated aggregates has previously been demonstrated,^{16,17} and is consistent with direction of these inclusions to autophagosomes for clearance via autophagy.¹⁸

5) The increased puncta formation of Atg8 and ref(2)P is not necessarily a sign of altered autophagy. The authors need to perform autophagy flux assays to make such claims.

We agree and now show impaired autophagic flux in flies lacking Appl function using the well-validated GFP-mCherry-Atg8a reporter (Supplementary Fig. 4e,f).¹⁹

6) Should overexpression of Atg8a not lead to clearance of ubiquitin-positive structures? It is not clear what process the increase in ubiquitin puncta upon Atg8a overexpression reflects. Is Atg8a co-aggregating, is it decorating perturbed endosomal membranes or is Atg8a helping to form autophagosomes? But if the latter is the case, why are ubiquitin puncta not reduced?

The reviewer raises another interesting point. While a prior study reported that increasing expression of Atg8a reduced biochemically insoluble, ubiquitinated protein,²⁰ others have not observed increased autophagy following overexpression of Atg8a/LC3,^{21,22} perhaps reflecting the need to coordinately regulate multiple steps downstream of Atg1 to induce effective autophagic degradation of client proteins. Alternatively, fine tuning of autophagic activity may be needed to maintain normal cellular proteostasis and health.²³ Consistent with our current findings,

increased numbers of protein aggregates has previously been observed following Atg8a overexpression in a separate aggregating protein disease model in flies.²⁴ A number of different mechanisms may contribute to Atg8a/LC3 overexpression toxicity with increased protein aggregation, including defective cargo recognition,²⁵ autophagy inhibition,²⁶ or altered membrane fusion.²⁷ We do not observe overexpressed Atg8a colocalizing with endosome markers (Supplementary Fig. 6e,f). We now address these important points in our discussion.

References

1. Fong, L. K. *et al.* Full-length amyloid precursor protein regulates lipoprotein metabolism and amyloid- β clearance in human astrocytes. *J Biol Chem* **293**, 11341–11357 (2018).
2. Davidsson, P., Bogdanovic, N., Lannfelt, L. & Blennow, K. Reduced expression of amyloid precursor protein, presenilin-1 and rab3a in cortical brain regions in Alzheimer's disease. *Dement Geriatr Cogn Disord* **12**, 243–250 (2001).
3. Johnston, J. A. *et al.* Quantification of APP and APLP2 mRNA in APOE genotyped Alzheimer's disease brains. *Brain Res Mol Brain Res* **43**, 85–95 (1996).
4. Colciaghi, F. *et al.* [alpha]-Secretase ADAM10 as well as [alpha]APPs is reduced in platelets and CSF of Alzheimer disease patients. *Mol Med* **8**, 67–74 (2002).
5. Ghosal, K. *et al.* Alzheimer's disease-like pathological features in transgenic mice expressing the APP intracellular domain. *Proc Natl Acad Sci U S A* **106**, 18367–18372 (2009).
6. Pulina, M. V., Hopkins, M., Haroutunian, V., Greengard, P. & Bustos, V. C99 selectively accumulates in vulnerable neurons in Alzheimer's disease. *Alzheimers Dement* **16**, 273–282 (2020).
7. Begcevic, I. *et al.* Brain-related proteins as potential CSF biomarkers of Alzheimer's disease: A targeted mass spectrometry approach. *J Proteomics* **182**, 12–20 (2018).
8. Habib, A., Sawmiller, D. & Tan, J. Restoring Soluble Amyloid Precursor Protein α Functions as a Potential Treatment for Alzheimer's Disease. *J Neurosci Res* **95**, 973–991 (2017).
9. Hashimoto, Y. *et al.* Transforming growth factor beta2 autocrinally mediates neuronal cell death induced by amyloid-beta. *J Neurosci Res* **83**, 1039–1047 (2006).
10. Bodmer, S., Podlisny, M. B., Selkoe, D. J., Heid, I. & Fontana, A. Transforming growth factor-beta bound to soluble derivatives of the beta amyloid precursor protein of Alzheimer's disease. *Biochem Biophys Res Commun* **171**, 890–897 (1990).
11. Penserga, T., Kudumala, S. R., Poulos, R. & Godenschwege, T. A. A Role for Drosophila Amyloid Precursor Protein in Retrograde Trafficking of L1-Type Cell Adhesion Molecule Neuroglian. *Front Cell Neurosci* **13**, 322 (2019).
12. Gems, D., Pletcher, S. & Partridge, L. Interpreting interactions between treatments that slow aging. *Aging Cell* **1**, 1–9 (2002).
13. Pérez-Pérez, J. M., Candela, H. & Micol, J. L. Understanding synergy in genetic interactions. *Trends Genet* **25**, 368–376 (2009).
14. Shen, D. *et al.* Novel cell- and tissue-based assays for detecting misfolded and aggregated protein accumulation within aggresomes and inclusion bodies. *Cell Biochem Biophys* **60**, 173–185 (2011).
15. Basisty, N. B. *et al.* Stable Isotope Labeling Reveals Novel Insights Into Ubiquitin-Mediated Protein Aggregation With Age, Calorie Restriction, and Rapamycin Treatment. *J Gerontol A Biol Sci Med Sci* **73**, 561–570 (2018).
16. Nezis, I. P. *et al.* Ref(2)P, the Drosophila melanogaster homologue of mammalian p62, is required for the formation of protein aggregates in adult brain. *J. Cell Biol.* **180**, 1065–1071 (2008).
17. Zirin, J., Nieuwenhuis, J., Samsonova, A., Tao, R. & Perrimon, N. Regulators of autophagosome formation in Drosophila muscles. *PLoS Genet* **11**, e1005006 (2015).

18. Lamark, T. & Johansen, T. Aggrephagy: selective disposal of protein aggregates by macroautophagy. *Int J Cell Biol* **2012**, 736905 (2012).
19. Kimura, S., Noda, T. & Yoshimori, T. Dissection of the autophagosome maturation process by a novel reporter protein, tandem fluorescent-tagged LC3. *Autophagy* **3**, 452–460 (2007).
20. Simonsen, A. *et al.* Promoting basal levels of autophagy in the nervous system enhances longevity and oxidant resistance in adult *Drosophila*. *Autophagy* **4**, 176–184 (2008).
21. Piracs, K. *et al.* Advantages and limitations of different p62-based assays for estimating autophagic activity in *Drosophila*. *PLoS ONE* **7**, e44214 (2012).
22. Maruzs, T., Simon-Vecsei, Z., Kiss, V., Csizmadia, T. & Juhász, G. On the Fly: Recent Progress on Autophagy and Aging in *Drosophila*. *Front Cell Dev Biol* **7**, 140 (2019).
23. Bjedov, I. *et al.* Fine-tuning autophagy maximises lifespan and is associated with changes in mitochondrial gene expression in *Drosophila*. *PLoS Genet* **16**, e1009083 (2020).
24. Sharma, A., Narasimha, K., Manjithaya, R. & Sheeba, V. Restoration of Sleep and Circadian Behavior by Autophagy Modulation in Huntington's Disease. *J Neurosci* **43**, 4907–4925 (2023).
25. Zaffagnini, G. *et al.* p62 filaments capture and present ubiquitinated cargos for autophagy. *EMBO J* **37**, e98308 (2018).
26. Wu, Z. *et al.* CRISPR/Cas9 Mediated GFP Knock-in at the MAP1LC3B Locus in 293FT Cells Is Better for Bona Fide Monitoring Cellular Autophagy. *Biotechnol J* **13**, e1700674 (2018).
27. Nakatogawa, H., Ichimura, Y. & Ohsumi, Y. Atg8, a ubiquitin-like protein required for autophagosome formation, mediates membrane tethering and hemifusion. *Cell* **130**, 165–178 (2007).

REVIEWERS' COMMENTS

Reviewer #1 (Remarks to the Author):

The authors addressed all my concerns and observations. In my opinion, the study is ready for publication.

Reviewer #3 (Remarks to the Author):

The authors have sufficiently answered all questions and concerns. No further revisions are requested.